# DyREP: Learning Representations over Dynamic Graphs

**Rakshit Trivedi**[1,*]**, Mehrdad Farajtabar**[2,*]**, Prasenjeet Biswal**[1] **& Hongyuan Zha**[1,*]
[1]Georgia Institute of Technology
[2]DeepMind

## Abstract

Representation Learning over graph structured data has received significant attention recently due to its ubiquitous applicability. However, most advancements have been made in static graph settings while efforts for jointly learning dynamic of the graph and dynamic on the graph are still in an infant stage. Two fundamental questions arise in learning over dynamic graphs: (i) How to elegantly model dynamical processes over graphs? (ii) How to leverage such a model to effectively encode evolving graph information into low-dimensional representations? We present **DyRep** - a novel modeling framework for dynamic graphs that posits representation learning as a *latent mediation process* bridging two observed processes namely – dynamics *of* the network (realized as topological evolution) and dynamics *on* the network (realized as activities between nodes). Concretely, we propose a two-time scale deep temporal point process model that captures the interleaved dynamics of the observed processes. This model is further parameterized by a temporal-attentive representation network that encodes temporally evolving structural information into node representations which in turn drives the nonlinear evolution of the observed graph dynamics. Our unified framework is trained using an efficient unsupervised procedure and has capability to generalize over unseen nodes. We demonstrate that DyRep outperforms state-of-the-art baselines for dynamic link prediction and time prediction tasks and present extensive qualitative insights into our framework.

## 1 Introduction

Representation learning over graph structured data has emerged as a keystone machine learning task due to its ubiquitous applicability in variety of domains such as social networks, bioinformatics, natural language processing, and relational knowledge bases. Learning node representations to effectively encode high-dimensional and non-Euclidean graph information is a challenging problem but recent advances in deep learning has helped important progress towards addressing it (Cao et al., 2015; Grover & Leskovec, 2016; Perozzi et al., 2014; Tang et al., 2015; Wang et al., 2016a; 2017; Xu et al., 2017), with majority of the approaches focusing on advancing the state-of-the-art in static graph setting. However, several domains now present highly dynamic data that exhibit complex temporal properties in addition to earlier cited challenges. For instance, social network communications, financial transaction graphs or longitudinal citation data contain fine-grained temporal information on nodes and edges that characterize the dynamic evolution of a graph and its properties over time.

These recent developments have created a conspicuous need for principled approaches to advance graph embedding techniques for dynamic graphs (Hamilton et al., 2017b). We focus on two pertinent questions fundamental to representation learning over dynamic graphs: (i) *What can serve as an elegant model for dynamic processes over graphs?* — A key modeling choice in existing representation learning techniques for dynamic graphs (Goyal et al., 2017; Zhou et al., 2018; Trivedi et al., 2017; Ngyuyen et al., 2018; Yu et al., 2018) assume that graph dynamics evolve as a single time scale process. In contrast to these approaches, we observe that most real-world graphs exhibit at least two distinct dynamic processes that evolve at different time scales — *Topological Evolution*: where the number of nodes and edges are expected to grow (or shrink) over time leading to structural changes in the graph; and *Node Interactions*: which relates to activities between nodes that may or may not be structurally connected. Modeling interleaved dependencies between these non-linearly evolving dynamic processes is a crucial next step for advancing the formal models of dynamic graphs.

---

*Corresponding Author: {rstrivedi@gatech.edu, farajtabar@google.com, zha@cc.gatech.edu}

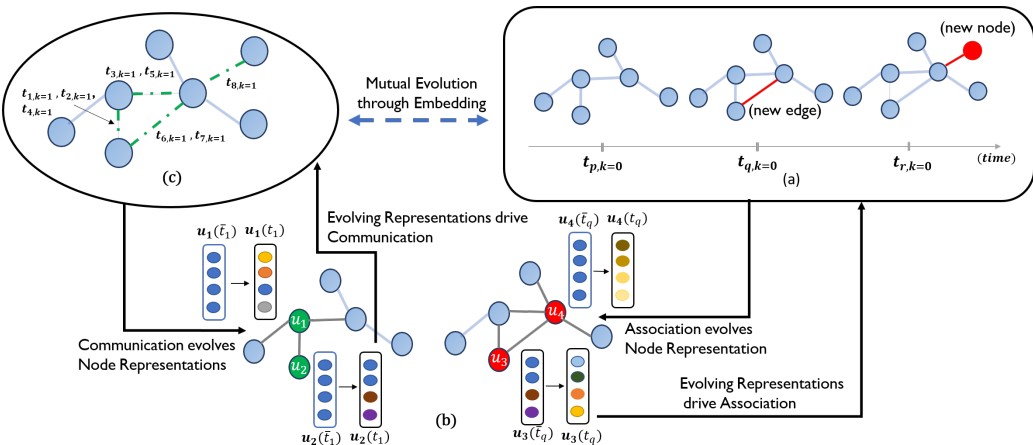

Figure 1: Evolution Through Mediation. **(a)** Association events (k=0) where the node or edge grows. **(c)** Communication Events (k=1) where nodes interact with each other. For both these processes, $t_{p,k=0} < (t_1, t_2, t_3, t_4, t_5)_{k=1} < t_{q,k=0} < (t_6, t_7)_{k=1} < t_{r,k=0}$. **(b)** Evolving Representations.

(ii) *How can one leverage such a model to learn dynamic node representations that are effectively able to capture evolving graph information over time?* — Existing techniques in this direction can be divided into two approaches: a.) Discrete-Time Approach, where the evolution of a dynamic graph is observed as collection of static graph snapshots over time (Zhu et al., 2016; Goyal et al., 2017; Zhou et al., 2018). These approaches tend to preserve (encode) very limited structural information and capture temporal information at a very coarse level which leads to loss of information between snapshots and lack of ability to capture fine-grained temporal dynamics. Another challenge in such approaches is the selection of appropriate aggregation granularity which is often misspecified. b.) Continuous-Time Approach, where evolution is modeled at finer time granularity in order to address the above challenges. While existing approaches have demonstrated to be very effective in specific settings, they either model simple structural and complex temporal properties in a decoupled fashion (Trivedi et al., 2017) or use simple temporal models (exponential family in (Ngyuyen et al., 2018)). But several domains exhibit highly nonlinear evolution of structural properties coupled with complex temporal dynamics and it remains an open problem to effectively model and learn informative representations capturing various dynamical properties of such complex systems.

As noted in (Chazelle, 2012), an important requirement to effectively learn over such dynamical systems is the ability to express the dynamical processes at different scales. We propose that any dynamic graph must be minimally expressed as a result of two fundamental processes evolving at different time scales: **Association Process** (dynamics *of* the network), that brings change in the graph structure and leads to long lasting information exchange between nodes; and **Communication Process** (dynamics *on* the network), that relates to activities between (not necessarily connected) nodes which leads to temporary information flow between them (Farine, 2017; Artime et al., 2017). We, then, posit our goal of learning node representations as modeling a *latent mediation process* that bridges the above two observed processes such that learned representations drive the complex temporal dynamics of both processes and these processes subsequently lead to the nonlinear evolution of node representations. Further, the information propagated across the graph is governed by the temporal dynamics of communication and association histories of nodes with its neighborhood. For instance, in a social network, when a node's neighborhood grows, it changes that node's representation which in turn affects her social interactions (*association* → **embedding** → *communication*). Similarly, when node's interaction behavior changes, it affects the representation of her neighbors and herself which in turn changes the structure and strength of her connections due to link addition or deletion (*communication* → **embedding** → *association*). We call this phenomenon — *evolution through mediation* and illustrate it graphically in Figure 1.

In this work, we propose a novel representation learning framework for dynamic graphs, **DyRep**, to model interleaved evolution of two observed processes through latent mediation process expressed above and effectively learn richer node representations over time. Our framework ingests dynamic graph information in the form of association and communication events over time and updates the node representations as they appear in these events. We build a two-time scale deep temporal point

process approach to capture the continuous-time fine-grained temporal dynamics of the two observed processes. We further parameterize the conditional intensity function of the temporal point process with a deep inductive representation network that learns functions to compute node representations. Finally, we couple the structural and temporal components of our framework by designing a novel *Temporal Attention Mechanism*, which induces temporal attentiveness over neighborhood nodes using the learned intensity function. This allows to capture highly interleaved and nonlinear dynamics governing node representations over time. We design an efficient unsupervised training procedure for end-to-end training of our framework. We demonstrate consistent and significant improvement over state-of-the-art representative baselines on two real-world dynamic graphs for the tasks of dynamic link prediction and time prediction. We further present an extensive qualitative analysis through embedding visualization and ablation studies to discern the effectiveness of our framework.

## 2 BACKGROUND AND PRELIMINARIES

### 2.1 RELATED WORK

Representation Learning approaches for static graphs either perform node embedding (Cao et al., 2015; Grover & Leskovec, 2016; Perozzi et al., 2014; Tang et al., 2015; Wang et al., 2016a; 2017; Xu et al., 2017) or sub-graph embedding (Scarselli et al., 2009; Li et al., 2016; Dai et al., 2016) which can also utilize convolutional neural networks (Kipf & Welling, 2017; 2016; Bruna et al., 2014). Among them, GraphSage (Hamilton et al., 2017a) is an inductive method for learning functions to compute node representations that can be generalized to unseen nodes. Most of these approaches only work with static graphs or can model evolving graphs without temporal information. Dynamic network embedding is pursued through various techniques such as matrix factorization (Zhu et al., 2016), structural properties (Zhou et al., 2018), CNN-based approaches (Seo et al., 2016), deep recurrent models (Trivedi et al., 2017), and random walks (Ngyuyen et al., 2018). There exists a rich body of literature on temporal modeling of dynamic networks (Kim et al., 2017), that focus on link prediction tasks but their goal is orthogonal to our work as they build task specific methods and do not focus on representation learning. Authors in (Yang et al., 2017; Sarkar et al., 2007) proposed models of learning dynamic embeddings but none of them consider time at finer level and do not capture both topological evolution and interactions simultaneously. In parallel, research on deep point process models include parametric approaches to learn intensity (Du et al., 2016; Mei & Eisner, 2017) using recurrent neural networks and GAN based approaches to learn intensity functions (Xiao et al., 2017). More detailed related works are provided in **Appendix F**.

### 2.2 TEMPORAL POINT PROCESSES

Stochastic point processes (Daley & Vere-Jones, 2007) are random processes whose realization comprises of discrete events in time, $t_1, t_2, \ldots$. A temporal point process is one such stochastic process that can be equivalently represented as a counting process, $N(t)$, which contains the number of events up to time $t$. The common way to characterize temporal point processes is via the conditional intensity function $\lambda(t)$, a stochastic model of rate of happening events given the previous events. Formally, $\lambda(t)\mathrm{d}t$ is the conditional probability of observing an event in the tiny window $[t, t + \mathrm{d}t)$, $\lambda(t)\mathrm{d}t := \mathbb{P}[\text{event in } [t, t + \mathrm{d}t)|\mathcal{T}(t)] = \mathbb{E}[\mathrm{d}N(t)|\mathcal{T}(t)]$, where $\mathcal{T}(t) = t_k|t_k < t$ is history until $t$. Similarly, for $t > t_n$ and given history $\mathcal{T} = t_1, \ldots, t_n$, we characterize the conditional probability that no event happens during $[t_n, t)$ as $S(t|\mathcal{T}) = \exp\left(-\int_{t_n}^t \lambda(\tau)\,\mathrm{d}\tau\right)$, which is called survival function of the process (Aalen et al., 2008). Moreover, the conditional density that an event occurs at time $t$ is defined as $f(t) = \lambda(t)\,S(t)$. The intensity $\lambda(t)$ is often designed to capture phenomena of interests – common forms include Poisson Process, Hawkes processes (Farajtabar et al., 2014; Hawkes, 1971; Wang et al., 2016b; Tabibian et al., 2017), Self-Correcting Process (Isham & Westcott, 1979). Temporal Point Processes have previously been used to model both – dynamics on the network (Farajtabar et al., 2016; Zarezade et al., 2017; Farajtabar et al., 2017) and dynamics of the network (Tran et al., 2015; Farajtabar et al., 2015).

### 2.3 NOTATIONS AND DYNAMIC GRAPH SETTING

**Notations.** Let $\mathcal{G}_t = (\mathcal{V}_t, \mathcal{E}_t)$ denote graph $\mathcal{G}$ at time $t$, where $\mathcal{V}_t$ is the set of nodes and $\mathcal{E}_t$ is the set of edges in $\mathcal{G}_t$ and the edges are undirected. ***Event Observation*** – Both communication and association processes are realized in the form of dyadic events observed between nodes on graph $\mathcal{G}$ over a temporal window $[t_0, T]$ and ordered by time. We use the following canonical tuple representation for any type of event at time $t$ of the form $e = (u, v, t, k)$, where $u, v$ are the two nodes involved

in an event. $t$ represents time of the event. $k \in \{0, 1\}$ and we use $k = 0$ to signify events from the topological evolution process (association) and $k = 1$ to signify events from node interaction process (communication). Persistent edges in the graph only appear through topological events while interaction events do not contribute them. Hence, $k$ represents an abstraction of scale (evolution rate) associated with processes that generate topological (dynamic of the network) and interaction events (dynamic on the network) respectively. We then represent complete set of $P$ observed events ordered by time in window $[0, T]$ as $\mathcal{O} = \{(u, v, t, k)_p\}_{p=1}^P$. Here, $t_p \in \mathbb{R}^+$, $0 \leq t_p \leq T$. **Appendix B** discusses a marked point process view of such an event set. ***Node Representation—*** Let $\mathbf{z}^v \in \mathbb{R}^d$ represent $d$-dimensional representation of node $v$. As the representation evolve over time, we qualify them as function of time: $\mathbf{z}^v(t)$ — the representation of node $v$ being updated after an event involving $v$ at time $t$. We use $\mathbf{z}^v(\bar{t})$ for most recently updated embedding of node $v$ just before $t$.

**Dynamic Graph Setting.** Let $\mathcal{G}_{t_0} = (\mathcal{V}_{t_0}, \mathcal{E}_{t_0})$ be the initial snapshot of a graph at time $t_0$. Please note that $\mathcal{G}_{t_0}$ may be empty or it may contain an initial structure (association edges) but it will not have any communication history. Our framework observes evolution of graph as a stream of events $\mathcal{O}$ and hence any new node will always be observed as a part of such an event. This will induce a natural ordering over nodes as available from the data. As our method is inductive, we never learn node-specific representations and rather learn functions to compute node representations. In this work, we only support growth of network i.e. we only model addition of nodes and structural edges and leave deletion as future work. Further, for general description of the model, we will assume that an edge in the graph do not have types and nodes do not have attributes but we discuss the details on how to use our model to accommodate these features in Appendix B.

## 3 PROPOSED METHOD: DYREP

The key idea of DyRep is to build a unified architecture that can ingest evolving information over graphs and effectively model the *evolution through mediation* phenomenon described in Section 1. To achieve this, we design a two-time scale temporal point process model of observed processes and parameterize it with an inductive representation network which subsequently models the latent mediation process of learning node representations. The rationale behind our framework is that the observed set of events are the realizations of the nonlinear dynamic processes governing the changes in topological structure of graph and interactions between the nodes in the graph. Now, when an event is observed between two nodes, information flows from the neighborhood of one node to the other and affects the representations of the nodes accordingly. While a communication event (interaction) only propagates local information across two nodes, an association event changes the topology and thereby has more global effect. The goal is to learn node representations that encode information evolving due to such local and global effects and further drive the dynamics of the observed events.

### 3.1 MODELING TWO-TIME SCALE OBSERVED GRAPH DYNAMICS

The observations over dynamic graph contain temporal point patterns of two interleaved complex processes in the form of communication and association events respectively. At any time $t$, the occurrence of an event, from either of these processes, is dependent on the most recent state of the graph, *i.e.*, two nodes will participate in any event based on their most current representations. Given an observed event $p = (u, v, t, k)$, we define a continuous-time deep model of temporal point process using the conditional intensity function $\lambda_k^{u,v}(t)$ that models the occurrence of event $p$ between nodes $u$ and $v$ at time $t$:

$$\lambda_k^{u,v}(t) = f_k(g_k^{u,v}(\bar{t})) \tag{1}$$

where $\bar{t}$ signifies the timepoint just before current event. The inner function $g_k(\bar{t})$ computes the compatibility of the most recently updated representations of two nodes, $\mathbf{z}^u(\bar{t})$ and $\mathbf{z}^v(\bar{t})$ as follows:

$$g_k^{u,v}(\bar{t}) = \boldsymbol{\omega}_k^T \cdot [\mathbf{z}^u(\bar{t}); \mathbf{z}^v(\bar{t})] \tag{2}$$

[;] signifies concatenation and $\boldsymbol{\omega}_k \in \mathbb{R}^{2d}$ serves as the model parameter that learns time-scale specific compatibility. $g_k(\bar{t})$ is a function of node representations learned through a representation network described in Section 3.2. This network parameterizes the intensity function of the point process model which serves as a unifying factor. Note that the dynamics are not two simple point processes dependent on each other, but, they are related through the mediation process and in the embedding space. Further, a well curated attention mechanism is employed to learn how the past drives future.

The choice of outer function $f_k$ needs to account for two critical criteria: 1) Intensity needs to be positive. 2) As mentioned before, the dynamics corresponding to communication and association

processes evolve at different time scales. To account for this, we use a modified version of softplus function parameterized by a dynamics parameter $\psi_k$ to capture this *timescale* dependence:

$$f_k(x) = \psi_k \log(1 + \exp(x/\psi_k)) \tag{3}$$

where, $x = g(\bar{t})$ in our case and $\psi_k(> 0)$ is scalar time-scale parameter learned as part of training. $\psi_k$ corresponds to the rate of events arising from a corresponding process. In 1D event sequences, the formulation in (3) corresponds to the nonlinear transfer function in (Mei & Eisner, 2017).

## 3.2 Learning latent Mediation Process Via Temporally Attentive Representation Network

We build a deep recurrent architecture that parameterizes the intensity function in Eq. (1) and learns functions to compute node representations. Specifically, after an event has occurred, the representation of both the participating nodes need to be updated to capture the effect of the observed event based on the principles of:

**Self-Propagation.** Self-propagation can be considered as a minimal component of the dynamics governing an individual node's evolution. A node evolves in the embedded space with respect to its previous position (e.g. set of features) and not in a random fashion.

**Exogenous Drive.** Some exogenous force may smoothly update the node's current features during the time interval (e.g. between two global events involving that node).

**Localized Embedding Propagation.** Two nodes involved in an event form a temporary (communication) or a permanent (association) pathway for the information to propagate from the neighborhood of one node to the other node. This corresponds to the influence of the nodes at second-order proximity passing through the other node participating in the event (See **Appendix A** for pictorial depiction).

To realize the above processes in our setting, we first describe an example setup: Consider nodes $u$ and $v$ participating in any type of event at time $t$. Let $\mathcal{N}_u$ and $\mathcal{N}_v$ denote the neighborhood of nodes $u$ and $v$ respectively. We discuss two key points here: 1) Node $u$ serves as a bridge passing information from $\mathcal{N}_u$ to node $v$ and hence $v$ receives the information in an aggregated form through $u$. 2) While each neighbor of $u$ passes its information to $v$, the information that node $u$ relays is governed by an aggregate function parametrized by $u$'s communication and association history with its neighbors.

With this setup, for any event at time $t$, we update the embeddings for both nodes involved in the event using a recurrent architecture. Specifically, for $p$-th event of node $v$, we evolve $\mathbf{z}^v$ as:

$$\mathbf{z}^v(t_p) = \sigma(\ \underbrace{\mathbf{W}^{struct}\mathbf{h}^u_{struct}(\bar{t_p})}_{\text{Localized Embedding Propagation}} + \underbrace{\mathbf{W}^{rec}\mathbf{z}^v(\bar{t^v_p})}_{\text{Self-Propagation}} + \underbrace{\mathbf{W}^t(t_p - \bar{t^v_p})}_{\text{Exogenous Drive}}), \tag{4}$$

where, $\mathbf{h}^u_{struct} \in \mathbb{R}^d$ is the output representation vectors obtained from aggregator function on node $u$'s neighborhood and $\mathbf{z}^v(\bar{t^v_p})$ is the recurrent state obtained from the previous representation of node $v$. $t_p$ is time point of current event, $\bar{t_p}$ signifies the timepoint just before current event and $\bar{t^v_p}$ represent time point of previous event for node $v$. $\mathbf{z}^v(\bar{t^v_p} = 0)$, the initial representation of a node $v$ may be initialized either using input node features from dataset or random vector as per the setting. Eq. 4 is a neural network based functional form parameterized by $\mathbf{W}^{struct}, \mathbf{W}^{rec} \in \mathbb{R}^{d \times d}$ and $\mathbf{W}^t \in \mathbb{R}^d$ that govern the aggregate effect of all the three inputs (graph structure, previous embedding and exogenous feature) respectively to compute representations. The above formulation is inductive (supports unseen nodes) and flexible (supports node and edge types) as discussed in **Appendix B**.

### 3.2.1 Temporally Attentive Aggregation

The Localized Embedding Propagation principle above captures rich structural properties based on neighborhood structure which is a key to any representation learning task over graphs. However, for a given node, not all of its neighbors are uniformly important and hence it becomes extremely important to capture information from each neighbor in some weighted fashion. Recently proposed attention mechanisms have shown great success in dealing with variable sized inputs, focusing on the most relevant parts of the input to make decisions. However, existing approaches consider attention as a static quantity. In dynamic graphs, changing neighborhood structure and interaction activities between nodes evolves importance of each neighbor to a node over time, thereby making attention itself a temporally evolving quantity. Further this quantity is dependent on the temporal history of association and communication of neighboring nodes through evolving representations. To this

---

**Algorithm 1** Update Algorithm for $\mathcal{S}$ and $\mathbf{A}$

---

**Input:** Event record $o = (u, v, t, k)$, Event Intensity $\lambda_k^{u,v}(t)$ computed in (1), most recently updated $\mathbf{A}(\bar{t})$ and $\mathcal{S}(\bar{t})$. **Output:** $\mathbf{A}(t)$ and $\mathcal{S}(t)$

**1.** Update $\mathbf{A}$ : $\mathbf{A}(t) = \mathbf{A}(\bar{t})$
**if** $k = 0$ **then** $\mathbf{A}_{uv}(t) = \mathbf{A}_{vu}(t) = 1$                ←{*Association event*}

**2.** Update $\mathcal{S}$ : $\mathcal{S}(t) = \mathcal{S}(\bar{t})$
**if** $k = 1$ and $\mathbf{A}_{uv}(t) = 0$ **return** $\mathcal{S}(t), \mathbf{A}(t)$     ←{*Communication event, no Association exists*}
**for** $j \in \{u, v\}$ **do**
    $b = \frac{1}{|\mathcal{N}_j(t)|}$ where $|\mathcal{N}_j(t)|$ is the size of $\mathcal{N}_j(t) = \{i : \mathbf{A}_{ij}(t) = 1\}$
    $\mathbf{y} \leftarrow \mathcal{S}_j(t)$
    **if** $k = 1$ and $\mathbf{A}_{uv}(t) = 1$ **then** {       ←{*Communication event, Association exists*}}
       $\mathbf{y}_i = b + \lambda_k^{ji}(t)$ where $i$ is the other node involved in the event.    ←{$\lambda$ *computed in Eq. 2*}
    **else if** $k = 0$ and $\mathbf{A}_{uv}(t) = 0$ **then** {             ←{*Association event*}}
       $b' = \frac{1}{|\mathcal{N}_j(\bar{t})|}$ where $|\mathcal{N}_j(\bar{t})|$ is the size of $\mathcal{N}_j(\bar{t}) = \{i : \mathbf{A}_{ij}(\bar{t}) = 1\}$
       $x = b' - b$
       $\mathbf{y}_i = b + \lambda_k^{ji}(t)$ where $i$ is the other node involved in the event    ←{$\lambda$ *computed in Eq. 2*}
       $\mathbf{y}_w = \mathbf{y}_w - x; \;\; \forall w \neq i, \; \mathbf{y}_w \neq 0$
    **end if**
    Normalize $\mathbf{y}$ and set $\mathcal{S}_j(t) \leftarrow \mathbf{y}$
**end for**
**return** $\mathcal{S}(t), \mathbf{A}(t)$

---

end, we propose a novel *Temporal Point Process based Attention Mechanism* that uses temporal information to compute the attention coefficient for a structural edge between nodes. These coefficient are then used to compute the aggregate quantity ($\mathbf{h_{struct}}$) required for embedding propagation.

Let $\mathbf{A}(t) \in \mathbb{R}^{n \times n}$ be the adjacency matrix for graph $\mathcal{G}_t$ at time $t$. Let $\mathcal{S}(t) \in \mathbb{R}^{n \times n}$ be a stochastic matrix capturing the strength between pair of vertices at time $t$. One can consider $\mathcal{S}$ as a selection matrix that induces a ***natural selection*** process for a node – it would tend to communicate more with other nodes that it wants to associate with or has recently associated with. And it would want to attend less to non-interesting nodes. We start with following implication required for the construction of $\mathbf{h}_{struct}^u$ in (4): For any two nodes $u$ and $v$ at time $t$, $\mathcal{S}_{uv}(t) \in [0, 1]$ if $\mathbf{A}_{uv}(t) = 1$ and $\mathcal{S}_{uv}(t) = 0$ if $\mathbf{A}_{uv}(t) = 0$. Denote $\mathcal{N}_u(t) = \{i : \mathbf{A}_{iu}(t) = 1\}$ as the 1-hop neighborhood of node $u$ at time $t$.

To formally capture the difference in the influence of different neighbors, we propose a novel *conditional intensity based attention layer* that uses the matrix $\mathcal{S}$ to induce a shared attention mechanism to compute attention coefficients over neighborhood. Specifically, we perform *localized attention* for a given node $u$ and compute the coefficients pertaining to the 1-hop neighbors $i$ of node $u$ as: $q_{ui}(t) = \frac{\exp(\mathcal{S}_{ui}(\bar{t}))}{\sum_{i' \in \mathcal{N}_u(t)} \exp(\mathcal{S}_{ui'}(\bar{t}))}$, where $q_{ui}$ signifies the attention weight for the neighbor $i$ at time $t$ and hence it is a temporally evolving quantity. These attention coefficients are then used to compute the aggregate information $\mathbf{h}_{struct}^u(\bar{t})$ for node $u$ by employing an attended aggregation mechanism across neighbors as follows: $\mathbf{h}_{struct}^u(\bar{t}) = \max \left( \left\{ \sigma \left( q_{ui}(t) \cdot \mathbf{h}^i(\bar{t}) \right), \forall i \in \mathcal{N}_u(\bar{t}) \right\} \right)$, where, $\mathbf{h}^i(\bar{t}) = \mathbf{W}^h \mathbf{z}^i(\bar{t}) + \mathbf{b}^h$ and $\mathbf{W}^h \in \mathbb{R}^{d \times d}$ and $\mathbf{b}^h \in \mathbb{R}^d$ are parameters governing the information propagated by each neighbor of $u$. $\mathbf{z}^i(\bar{t}) \in \mathbb{R}^d$ is the most recent embedding for node $i$. The use of $\max$ operator is inspired from learning on general point sets (Qi et al., 2017). By applying max-pooling operator element-wise, the model effectively captures different aspects of the neighborhood. We found $\max$ to work slightly better as it considers temporal aspect of neighborhood which would be amortized if $mean$ is used instead.

**Connection to Neural Attention over Graphs.** Our proposed temporal attention layer shares the motivation of recently proposed Graph Attention Networks (GAT) (Veličković et al., 2018) and Gated Attention Networks (GaAN) (Zhang et al., 2018) in the spirit of applying non-uniform attention over neighborhood. Both GAT and GaAN have demonstrated significant success in static graph setting. GAT advances GraphSage (Hamilton et al., 2017a) by employing multi-head non-uniform attention

over neighborhood and GaAN advances GAT by applying different weights to different heads in the multi-head attention formulation. The key innovation in our model is the parameterization of attention mechanism by a point process based temporal quantity $\mathcal{S}$ that is evolving and drives the impact that each neighbor has on the given node. Further, unlike static methods, we use these attention coefficients as input to the aggregator function for computing the temporal-structural effect of neighborhood. Finally, static methods use multi-head attention to stabilize learning by capturing multiple representation spaces but this is an inherent property in our layer as representations and event intensities update over time and hence new events help capture multiple representation spaces.

**Construction and Update of $\mathcal{S}$.** We construct a single stochastic matrix $\mathcal{S}$ (used to parameterize attention in the earlier section) to capture complex temporal information. At the initial timepoint $t = t_0$, we construct $\mathcal{S}(t_0)$ directly from $\mathbf{A}(t_0)$. Specifically, for a given node $v$, we initialize the elements of corresponding row vector $\mathcal{S}_v(t_0)$ as: $\mathcal{S}_{vu}(t_0) = 0$ if ($v = u$ or $\mathbf{A}_{vu}(t_0) = 0$) and $\mathcal{S}_{vu}(t_0) = \frac{1}{|\mathcal{N}_v(t_0)|}$ if $\mathcal{N}_v(t_0) = \{u : \mathbf{A}_{uv}(t_0) = 1\}$. After observing an event $o = (u, v, t, k)$ at time $t > t_0$, we make updates to $\mathbf{A}$ and $\mathcal{S}$ as per the observation of $k$. Specifically, $\mathbf{A}$ only gets updated for association events (k=0, change in structure). Note that $\mathcal{S}$ is parameter for a structural temporal attention which means temporal attention is only applied on structural neighborhood of a node. Hence, the values of $\mathcal{S}$ are only updated/active in two scenarios: a) the current event is an interaction between nodes which already has structural edge ($\mathbf{A}_{uv}(t) = 1$ and $k = 1$) and b) the current event is an association event ($k = 0$). Given a neighborhood of node $u$, $b$ represents background (base) attention for each edge which is uniform attention based on neighborhood size. Whenever an event involving $u$ occurs, this attention changes in following ways: For case (a), the attention value for corresponding $\mathcal{S}$ entries are updated using the intensity of the event. For case (b), repeat same as (a) but also adjust the background attention (by $b - b'$, $b$ and $b'$ being the new and old background attention respectively) for edge with other neighbors as the neighborhood size grows in this case. From mathematical viewpoint, this update resembles a standard temporal point process formulation where the term coming from $b$ serves as background attention while $\lambda$ can be viewed as endogenous intensity based attention. Algorithm 1 outlines complete update scenarios. In the directed graph case, updates to $\mathbf{A}$ will not be symmetric, which will subsequently affect the neighborhood structure and attention flow for a node. **Appendix A** provides a pictorial depiction of the complete DyRep framework discussed in this section. We provide an extensive ablation study in **Appendix C** that can help discern the contribution of all the above components in achieving our goal.

## 4 EFFICIENT LEARNING PROCEDURE

The complete parameter space for the current model is $\mathbf{\Omega} = \{\mathbf{W}^{struct}, \mathbf{W}^{rec}, \mathbf{W}^t, \mathbf{W}^h, \mathbf{b}^h,$ $\{\boldsymbol{\omega}_k\}_{k=0,1}, \{\psi_k\}_{k=0,1}\}$. For a set $\mathcal{O}$ of $P$ observed events, we learn these parameters by minimizing the negative log likelihood: $\mathcal{L} = -\sum_{p=1}^{P} \log(\lambda_p(t)) + \int_0^T \Lambda(\tau)d\tau$, where $\lambda_p(t) = \lambda_{k_p}^{u_p, v_p}(t)$ represent the intensity of event at time $t$ and $\Lambda(\tau) = \sum_{u=1}^{n} \sum_{v=1}^{n} \sum_{k \in \{0,1\}} \lambda_k^{u,v}(\tau)$ represent total survival probability for events that do not happen. While it is intractable (will require $\mathcal{O}(n^2 k)$ time) and unnecessary to compute the integral in the log-likelihood equation for all possible non-events in a stochastic setting, we can locally optimize $\mathcal{L}$ using mini-batch stochastic gradient descent where we estimate the integral using novel sampling technique. Algorithm 2 in **Appendix H** adopts a simple variant of Monte Carlo trick to compute the survival term of log-likelihood equation. Specifically, in each mini-batch, we sample non-events instead of considering all pairs of non-events (which can be millions). Let $m$ be the mini-batch size and $N$ be the number of samples. The complexity of Algorithm 2 will then be $\mathcal{O}(2mkN)$ for the batch where the factor of 2 accounts for the update happening for two nodes per event which demonstrates linear scalability in number of events which is desired to tackle web-scale dynamic networks (Paranjape et al., 2017). The overall training procedure is adopted from (Trivedi et al., 2017) where the Backpropagation Through Time (BPTT) training is conducted over a global sequence, thereby maintaining the dependencies between events across sequences while avoiding gradient related issues. Implementation details are left to **Appendix G**.

## 5 EXPERIMENTS

### 5.1 DATASETS

We evaluate DyRep and baselines on two real world datasets: **Social Evolution Dataset** released by MIT Human Dynamics Lab — #nodes: 83, #Initial Associations: 376, #Final Associations: 791, #Communications: 2016339 and Clustering Coefficient: 0.548. **Github Dataset** available at Github Archive — #nodes: 12328, #Initial Associations: 70640, #Final Associations: 166565,

Table 1: Comparison of DyRep with state-of-the-art approaches

| Key Properties | DyRep (Our Method) | Know-Evolve (Dynamic) | DynGem (Dynamic) | GraphSage (Static) | GAT (Static) |
|---|---|---|---|---|---|
| Models Association | ✓ | X | ✓ | ✓ | ✓ |
| Models Communication | ✓ | ✓ | X | X | X |
| Models Time | ✓ | ✓ | X | X | X |
| Learns Representation | ✓ | ✓ | ✓ | ✓ | ✓ |
| Predicts Time | ✓ | ✓ | X | X | X |
| Graph Information | 2nd-order Neighborhood | Single Edge | 1st and 2nd-order Neighborhood | 2nd-order Neighborhood | 1st-order Neighborhood |
| Attention Mechanism | Temporal Point Process (Non-Uniform) | None | None | Sampling (Uniform) | Multi-head (Non-Uniform) |
| Learning | Unsupervised | Unsupervised | Semi-Supervised | Unsupervised | Supervised |

#Communications: 604649 and Clustering Coefficient: 0.087. These datasets cover a range of configurations as Social Dataset is a small network with high clustering coefficient and over 2M events. In contrast, Github dataset forms a large network with low clustering coefficient and sparse events thus allowing us to test the robustness of our model. Further, Github dataset contains several unseen nodes which were never encountered during training.

## 5.2 TASKS AND METRICS
We study the effectiveness of DyRep by evaluating our model on tasks of dynamic link prediction and event time prediction tasks:

**Dynamic Link Prediction.** When any two nodes in a graph has increased rate of interaction events, they are more likely to get involved in further interactions and eventually these interactions may lead to the formation of structural link between them. Similarly, formation of the structural link may lead to increased likelihood of interactions between newly connected nodes. To understand, how well our model captures these phenomenon, we ask questions like: Which is the most likely node $u$ that would undergo an event with a given node $v$ governed by dynamics $k$ at time $t$? The conditional density of such and event at time $t$ can be computed: $f_k^{u,v}(t) = \lambda_k^{u,v}(t) \cdot \exp\left(\int_{\bar{t}}^{t} \lambda(s)ds\right)$, where $\bar{t}$ is the time of the most recent event on either dimension $u$ or $v$. We use this conditional density to find most likely node.

For a given test record $(u, v, t, k)$, we replace $v$ with other entities in the graph and compute the density as above. We then rank all the entities in descending order of the density and report the rank of the ground truth entity. Please note that the latest embeddings of the nodes update even during the test while the parameters of the model remaining fixed. Hence, when ranking the entities, we remove any entities that creates a pair already seen in the test. We report Mean Average Rank (MAR) and HITS(@10) metric for dynamic link prediction.

**Event Time Prediction.** This is a relatively novel application where the aim is to compute the next time point when a particular type of event (structural or interaction) can occur. Given a pair of nodes $(u, v)$ and event type $k$ at time $t$, we use the above density formulation to compute conditional density at time $t$. The next time point $\hat{t}$ for the event can then be computed as: $\hat{t} = \int_{t}^{\infty} t f_k^{u,v}(t)dt$ where the integral does not have an analytic form and hence we estimate it using Monte Carlo trick. For a given test record $(u, v, t, k)$, we compute the next time this communication event may occur and report Mean Absolute Error (MAE) against the ground truth.

## 5.3 BASELINES
**Dynamic Link Prediction.** We compare the performance of our model against multiple representation learning baselines, four of which has capability to model evolving graphs. Specifically, we compare with **Know-Evolve** (Trivedi et al., 2017)— a state-of-the-art model for multi-relational dynamic graphs where each edge has time-stamp and type (communication events), **DynGem** (Goyal et al., 2017)—divides timeline into discrete time points and learns embedding for the graph snapshots at these time points. **DynTrd** (Zhou et al., 2018) focuses on specific structure of triad to model how close triads are formed from open triads in dynamic networks. **GraphSage** (Hamilton et al., 2017a)— an inductive representation learning method that learns sample and aggregation functions to learn representations instead of training for individual node. **Node2Vec** (Grover & Leskovec, 2016)—simple transductive baseline to learn graph embeddings over static graphs. Table 1 provides qualitative comparison between state-of-the-art methods and our framework. In our experiments, we compare with GraphSage instead of GAT as we share the unsupervised setting with GraphSage while

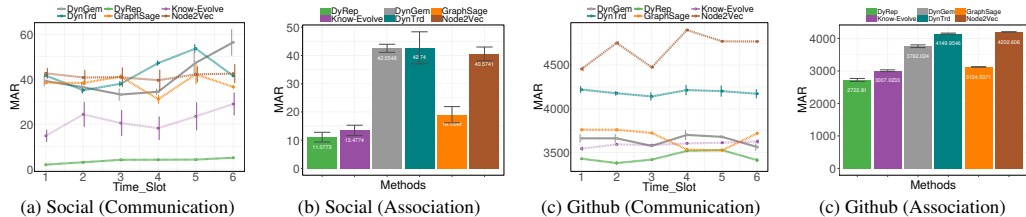

Figure 2: Dynamic Link Prediction Performance for **(a-b)** Social Evolution Dataset **(c-d)** Github Dataset. We report HITS@10 results and zoomed versions in **Appendix E**. Best viewed in pdf.

GAT is designed for supervised learning. In Appendix A (Ablation studies), we show results on one version where we only update attention based on Association events which is temporal analogous to GAT.

**Event Time Prediction.** We compare our model against (i) Know-Evolve which has the ability to predict time in a multi-relational dynamic graphs (II) Multi-dimensional Hawkes Process (MHP) (Du et al., 2015) model where all events in graph are considered as dyadic.

## 5.4 EVALUATION SCHEME

We divide our test sets into $n(=6)$ slots based on time and report the performance for each time slot, thus providing comprehensive temporal evaluation of different methods. This method of reporting is expected to provide fine-grained insights on how various methods perform over time as they move farther from the learned training history. For dynamic baselines that do not explicitly model time (DynGem, DynTrd, GraphSage) and static baselines (Node2Vec), we adopt a sliding window training approach with warm-start method where we learn on initial train set and test for the first slot. Then we add the data from first slot in the train set and remove equal amount of data from start of train set and retrain the model using the embeddings from previous train.

## 5.5 EXPERIMENTAL RESULTS

**Communication Event Prediction Performance.** We first consider the task of predicting communication events between nodes which may or may not have a permanent edge (association) between them. Figure 2 **(a-b)** shows corresponding results.

**Social Evolution.** Our method significantly and consistently outperforms all the baselines on both metrics. While the performance of our method drops a little over time, it is expected due to the temporal recency affect on node's evolution. Know-Evolve can capture event dynamics well and shows consistently better rank than others but its performance deteriorates significantly in HITS@10 metric over time. We conjecture that features learned through edge-level modeling limits the predictive capacity of the method over time. The inability of DynGem (snapshot based dynamic), DynTrd and GraphSage (inductive) to significantly outperform Node2vec (transductive static baseline) demonstrate that discrete time snapshot based models fail to capture fine-grained dynamics of communication events.

**Github dataset.** We demonstrate comparable performance with both Know-Evolve and GraphSage on Rank metric. We would like to note that overall performance for all methods on rank metric is low. As we reported earlier, Github dataset is very sparse with very low clustering coefficient which makes it a challenging dataset to learn. It is expected that for a large number of nodes with no communication history, most of the methods will show comparable performance but our method outperforms all others when there is some history available. This is demonstrated by our significantly better performance for HITS@10 metric where we are able to do highly accurate prediction for nodes where we learn better history. This can also be attributed to our model's ability to capture the effect of evolving topology which is missed by Know-Evolve. Finally, we do not see significant decrease in performance of any method over time in this case which can again be attributed to roughly uniform distribution of nodes with no communication history across time slots.

**Association Event Prediction Performance.** Association events are not available for all time slots so Figure 2 **(c-d)** report the aggregate number for this task. For both the datasets, our model significantly outperforms the baselines for this task. Specifically, our model's strong performance on HITS@10 metric across both datasets demonstrates its robustness in accurate learning from various properties of data. On Social evolution dataset, the number of association events are very small (only 485) and hence our strong performance shows that the model is able to capture the influence

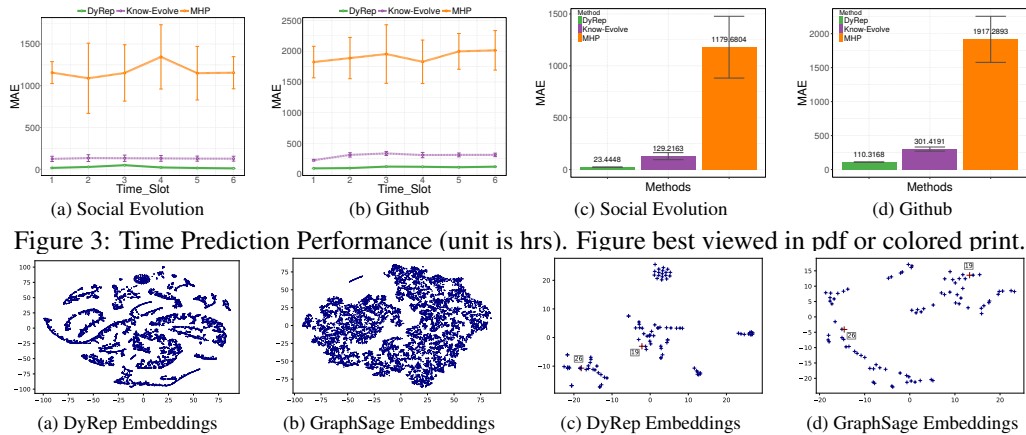

Figure 3: Time Prediction Performance (unit is hrs). Figure best viewed in pdf or colored print.

(a) DyRep Embeddings    (b) GraphSage Embeddings    (c) DyRep Embeddings    (d) GraphSage Embeddings

Figure 4: tSNE for learned embeddings after training. Figure best viewed in color.

of communication events on the association events through the learned representations (mediation). On the Github dataset, the network grows through new nodes and our model's strong performance across both metric demonstrates its inductive ability to generalize across new nodes across time. An interesting observation was poor performance of DynTrd which seems to be due to its objective to complete triangles. Github dataset is very sparse and has very few possibilities for triadic closure.

**Time Prediction Performance.** Figure 3 demonstrates consistently better performance than state-of-the-art baseline for event time prediction on both datasets. While Know-Evolve models both processes as two different relations between entities, it does not explicitly capture the variance in the time scales of two processes. Further, Know-Evolve does not consider influence of neighborhood which may lead to capturing weaker temporal-structural dynamics across the graph. MHP uses specific parametric intensity function which fails to account for intricate dependencies across graph.

**Qualitative Performance.** We conducted a series of qualitative analysis to understand the discriminative power of evolving embeddings learned by DyRep. We compare our embeddings against GraphSage embeddings as it is state-of-the-art embedding method that is also inductive. Figure 4 **(a-b)** shows the tSNE embeddings learned by Dyrep (left) and GraphSage (right) respectively. The visualization demonstrates that DyRep embeddings have more discriminative power as it can effectively capture the distinctive and evolving structural features over time as aligned with empirical evidence. Figure 4 **(c-d)** shows use case of two associated nodes (19 and 26) that has persistent edge but less communication for above two methods. DyRep keeps the embeddings nearby although not in same cluster (cos. dist. - 0.649) which demonstrates its ability to learn the association and less communication dynamics between two nodes. For GraphSage the embeddings are on opposite ends of cluster with (cos. dist. - 1.964). We provide more extensive analysis in **Appendix D**.

## 6 CONCLUSION

We introduced a novel modeling framework for dynamic graphs that effectively and efficiently learns node representations by posing representation learning as latent mediation process bridging dynamic processes of topological evolution and node interactions. We proposed a deep temporal point process model parameterized by temporally attentive representation network that models these complex and nonlinearly evolving dynamic processes and learns to encode structural-temporal information over graph into low dimensional representations. Our superior evaluation performance demonstrates the effectiveness of our approach compared to state-of-the-art methods. We present this work as the first generic and unified representation learning framework that adopts a novel modeling paradigm for dynamic graphs and support wide range of dynamic graph characteristics which can potentially have many exciting adaptations. As a part of our framework, we also propose a novel temporal point process based attention mechanism that can attend over neighborhood based on the history of communications and association events in the graph. Currently, DyRep does not support network shrinkage due to following reasons: (i) It is difficult to procure data with fine grained deletion time stamps and (ii) The temporal point process model requires more sophistication to support deletion. For example, one can augment the model with a survival process formulation to account for lack of node/edge at future time. Another interesting future direction could be to support encoding higher order dynamic structures.

## ACKNOWLEDGEMENTS

We sincerely thank our anonymous ICLR reviewers for critical feedback that helped us to improve the clarity and precision of our presentation. We would also like to thank Jiachen Yang (Georgia Tech) for insightful comments on improving the presentation of the paper. This work was supported in part by NSF IIS-1717916, NSF CMMI-1745382, NSFC-Zhejiang Joint Fund for the Integration of Industrialization and Information (U1609220) and National Science Foundation of China (61672231).

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

APPENDIX

# A    PICTORIAL EXPOSITION OF DYREP REPRESENTATION NETWORK

## A.1    LOCALIZED EMBEDDING PROPAGATION

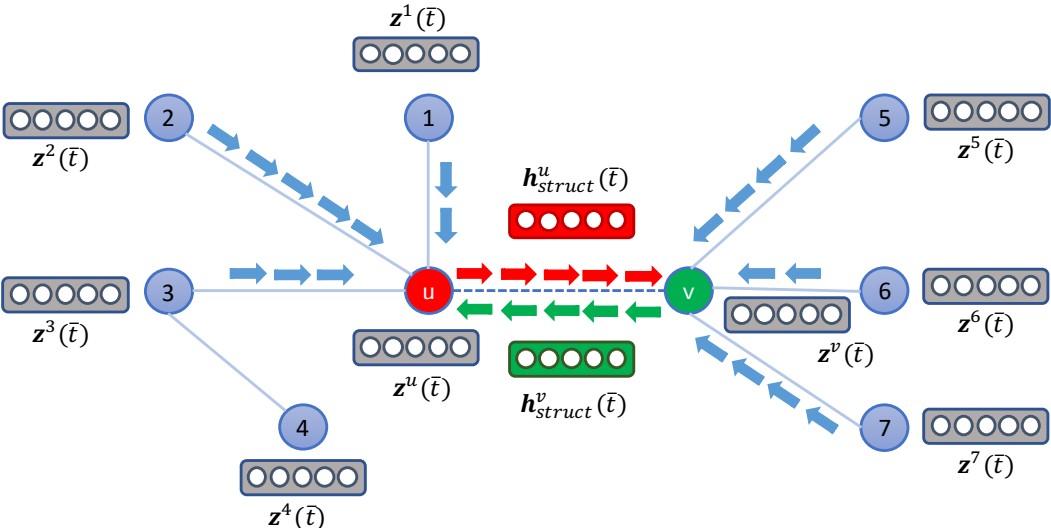

Figure 5: **Localized Embedding Propagation**: An event is observed between nodes $u$ and $v$ and $k$ can be 0 or 1 i.e. It can either be a topological event or interaction event. The first term in Eq 4. contains $\mathbf{h}_{struct}$ which is computed for updating each node involved in the event. For node $u$, the update will come from $\mathbf{h}^{v}_{struct}$ (green flow) and for node $v$, the update will come from $\mathbf{h}^{u}_{struct}$ (red flow). Please note all embeddings are dynamically evolving hence the information flow after every event is different and evolves in a complex fashion. With this mechanism, the information is passed from neighbors of node $u$ to node $v$ and neighbors of node $v$ to node $u$. (i) Interaction events lead to temporary pathway - such events can occur between nodes which are not connected. In that case, this flow will occur only once but it will not make $u$ and $v$ neighbors of each other (e.g. meeting at a conference). (ii) Topological events lead to permanent pathway - in this case $u$ and $v$ becomes neighbor of each other and hence will contribute to structural properties moving forward (e.g. being academic friends). The difference in number of blue arrows on each side signify different importance of each node to node $u$ and node $v$ respectively.

**Overall Embedding Update Process.** As a starting point, *neighborhood* only includes nodes connected by a structural edge. On observing an event, we update the embeddings of two nodes involved in the event using Eq 4. For a node $u$, the first term of Eq 4 (**Localized Embedding Propagation**) requires $\mathbf{h}_{struct}$ which is the information that is passed from neighborhood ($N_v$) of node $v$ to node $u$ via node $v$ (one can visualize $v$ as being the message passer from its neighborhood to $u$). This information is used to update the embedding of node $u$. However, we posit that node $v$ does not relay equal amount of information from its neighbors to node $u$. Rather, node $v$ receives its information to be relayed based on its communication and association history with its neighbors (which relates to importance of each neighbor). This requires to compute the attention coefficients on the structural edges between node $v$ and its neighbors. For any edge, we want this coefficient to be dependent on rate of events between the two nodes (thereby emulating real world phenomenon that one gains more information from people one interacts more with). Hence, we parameterize our attention module with the temporal point process parameter $\mathcal{S}_{uv}$. Algorithm 1 outlines the process of computing the value of this parameter.

## A.2 Computing $\mathbf{h}_{struct}$: Temporal Point Process based Attention

$\mathbf{z}^1(\bar{t})$

$\mathbf{z}^2(\bar{t})$

$q_{u2}(\bar{t})$

$q_{u1}(\bar{t})$

$\boldsymbol{h}^u_{struct}(\bar{t})$

$q_{u3}(\bar{t})$

$\mathbf{z}^3(\bar{t})$

$\mathbf{z}^u(\bar{t})$

$\mathbf{z}^v(\bar{t})$

$\mathbf{z}^4(\bar{t})$

Temporal Point Process Self-Attention:

$\boldsymbol{h}^u_{struct}(\bar{t})$
$= \max(\{\sigma(q_{ui}(\bar{t}) * \boldsymbol{h}^i(\bar{t}))\})$

$\boldsymbol{h}^i(\bar{t}) = \boldsymbol{W}^h \boldsymbol{z}^i(\bar{t}) + b^h$

where $i \in N_u(\bar{t})$ is the node in neighborhood of node u.

$q_{ui}(\bar{t})$
$= \dfrac{\exp(S_{ui}(\bar{t}))}{\sum_{i' \in N_u(\bar{t})} \exp(S_{ui'}(\bar{t}))}$

Figure 6: Temporal Point Process based Self-Attention: This figure illustrates the computation of $\mathbf{h}^u_{struct}$ for node $u$ to pass to node $v$ for the same event described before between nodes $u$ and $v$ at time $t$ with any $k$. $\mathbf{h}^u_{struct}$ is computed by aggregating information from neighbors (1,2,3) of $u$. However, Nodes that are closely connected or has higher interactions tend to attend more to each other compared to nodes that are not connected or nodes between which interactions is less even in presence of connection. Further, every node has a specific attention span for other node and therefore attention itself is a temporally evolving quantity. DyRep computes the temporally evolving attention based on association and communication history between connected nodes. The attention coefficient function ($q$'s) is parameterized by $\mathcal{S}$ which is computed using the intensity of events between connected nodes. Such attention mechanism allows the evolution of importance of neighbors to a particular node ($u$ in this case) which aligns with real-world phenomenon.

## A.3 Computing $\mathcal{S}$: Algorithm 1

Please check Figure 7 on next page.

## B Rationale Behind DyRep Framework

**Connection to Marked Point Process.** From a mathematical viewpoint, for any event $e$ at time $t$, any information other than the time point can be considered a part of mark space describing the events. Hence, for DyRep, given a one-dimensional timeline, one can consider $\mathcal{O} = \{(u, v, k)_p, t_p)_{p=1}^P$ as a marked process with the triple $(u, v, k)$ representing the mark.

However, from machine learning perspective, using a single-dimensional process with such marks does not allow to efficiently and effectively discover or model the structure in the point process useful for learning intricate dependencies between events, participants of the events and dynamics governing those events. Hence, it is often important to extract the information out of the mark space and build an abstraction that helps to *discover the structure* in point process and make this learning *parameter efficient*. In our case, this translates to two components:

1. The nodes in the graph are considered as dimensions of the point process, thus making it a multi-dimensional point process where an event represents interaction/structure between the dimensions, thus allowing us to explicitly capture dependencies between nodes.

2. The topological evolution of networks happen at much different temporal scale than activities on a fixed topology network (e.g. rate of making friends vs liking a post on a social network). However both these processes affect each other's evolution in a complex and nonlinear

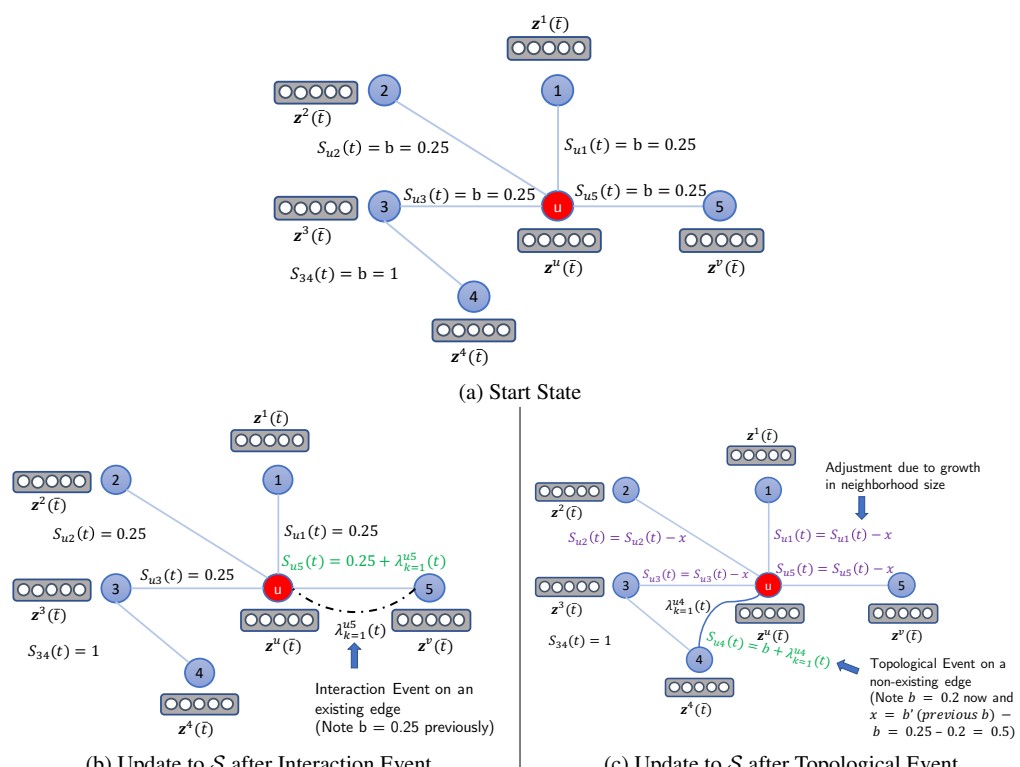

(a) Start State

(b) Update to $\mathcal{S}$ after Interaction Event

(c) Update to $\mathcal{S}$ after Topological Event

Figure 7: Computing $\mathcal{S}$. Illustration of the update to $\mathcal{S}$ under two circumstances for events that involve node $u$: (i) Interaction events between neighbors (ii) Topological Event between non-neighbors. We only illustrate one node but update will happen for both nodes in the event (e.g. for $(u,v)$, rows of both nodes will be updated asymmetrically due to different neighborhood size. **(a)** shows the initial state where $u$ has 4 neighbors and hence background attention is uniform $b = 0.25$. **(b)** $u$ has an interaction event with node 5. Update only happens to $\mathcal{S}_{u5}$ and $\mathcal{S}_{5u}$ based on intensity of the event. **(c)** $u$ has a topological event with node 4. $b$ changes to 0.2. $b' = 0.25$ which is the previous $b$. Update happens to $\mathcal{S}_{u4}$ and $\mathcal{S}_{4u}$ based on intensity of event. Next attention for all other neighbors of both nodes (We only show for $u$ here) are adjusted to reflect neighborhood size change. The matrix $\mathcal{S}$ is used for computing attention and hence does not get updated for interaction events between nodes which do not have an edge (for e.g. pair $(1,2)$ may have an interaction event $\mathcal{S}_{12}$ won't be updated as they are not neighbors.

fashion. Abstracting $k$ to associate it with these different scales of evolution facilitates to model our purpose of expressing dynamic graphs at two time scales in a principled manner. It also provides an ability to explicitly capture the influential dynamics (Chazelle, 2012) of topological evolution on dynamics of network activities and vice versa (through the learned embedding – aka **evolution through mediation**.

Note that this distinction in use of mark information is also important as we learn representations for nodes (dimensions) but not for $k$. It is important to realize that $k$ representing two different scales of event dynamics is not same as edge or interaction type. For instance, in case of typed persistent edge (e.g. wasbornIn, livesIn) or typed interaction (e.g. visit, fight), one would add type as another component in the mark space to represent an event while $k$ still signifying different dynamic scales.

**Comparison to (Trivedi et al., 2017).** In the similar vein as above, the point process specification of (Trivedi et al., 2017) can also be considered as a marked process that models the typed interaction dynamics at a single time-scale and does not model topological evolution. In contrast to that, our method explicitly models dynamic graph process at two time scales. While both models use a point process based formulation for modeling temporal dynamics, there are several significant methodological differences between the two approaches:

*Deep Point Process Model* — While one can augment the event specification in (Trivedi et. al.

2017) with additional mark information, that itself is not adequate to achieve DyRep's modeling of dynamical process over graphs at multiple time scales. We employ a softplus function for $f_k$ which contains a dynamic specific scale parameter $\psi_k$ to achieve this while (Trivedi et al. 2017) uses an exponential ($\exp$) function for $f$ with no scale parameter. Their intensity formulation attains a Rayleigh distribution which leads to a specific assumption about underlying dynamics which models fads where intensity of events drop rapidly between events after increasing. Our two-time scale model is more general and induces modularization, where each of two components allow complex, nonlinear and dependent dynamics towards a non-zero steady state intensity.

*Graph Structure*— As shown in (Hamilton et al., 2017b), the key idea behind representation learning over graphs is to capture both the global position and local neighborhood structural information of node into its representations. Hence, there has been significant research efforts invested in devising methods to incorporate graph structure into the computation of node representation. Aligned with these efforts, DyRep proposes a novel and sophisticated Localized Embedding Propagation principle that dynamically incorporates graph structure from both local neighborhood and faraway nodes (as interactions are allowed between nodes that do not have an edge). Contrary to that, (Trivedi et al., 2017) uses single edge level information, specific to the relational setting, into their representations.

*Deep Temporal Point Process Based Self-Attention*— For learning over graphs, attention has been shown to be extremely valuable as importance of nodes differ significantly relative to each other. The state-of-the-art approaches have focused solely on static graphs with Graph Attention Networks (Veličković et al., 2018) being the most recent one. Our attention mechanism for dynamic graphs present a significant and principled advancement over the existing state-of-the-art Graph based Neural Self-Attention techniques which only support static graphs. As (Trivedi et al., 2017) do not incorporate graph structure, they do not use any kind of attention mechanism.

**Support for Node Attributes and Edge Types.** Node types or attributes are supported in our work. In Eq. 4, $z^v(\bar{t}_p^{\,v})$ induces recurrence on node $v$'s embedding, but when node $v$ is observed for first time, $z^v(\bar{t}_p^{\,v}) = \mathbf{x}_v$ where $\mathbf{x}_v$ is randomly initialized or contains the raw node features available in data (which also includes type). One can also add an extra term in Eq. 4 to support high-dimension node attributes. Further, we also support different types of edges. If either the structural edge or an interaction has a type associated with it, our model can trivially support it in Eq. 3 and Eq. 4, first term $\mathbf{h}_{struct}$. Currently, for computing $\mathbf{h}_{struct}$, the formulation is shown to use aggregation over nodes. However, this aggregation can be augmented with edge type information as conventionally done in many representation learning frameworks (Hamilton et al., 2017b). Further, for more direct effect, Eq 3 can include edge type as third feature vector in the concatenation for computing $g_k$.

**Support for new nodes.** As mentioned in Section 2.3 of the main paper, the data contains a set of dyadic events ordered in time. Hence, each event involves two nodes $u$ and $v$. A new node will always appear as a part of such an event. Now, as mentioned above, the initial embedding of any new node $u$ is given by $z^u(\bar{t}_p^{\,u})$ which can be randomly initialized or using the raw feature vector of the node $u$, $\mathbf{x}_u$. This allows the computation of intensity function for the event involving new node in Eq 1. Due to the inductive ability of our framework, we can then compute the embedding of the new node using Eq 4. There are two cases possible: Either one of the two nodes are new or both nodes are new. The mechanism for these two cases work as follows:

- *Only one new node in observed event* — To compute the embedding of new nodes, $\mathbf{h}_{struct}$ is computed using neighborhood of the existing (other) node, $\mathbf{z}(t_0^u)$ s the feature vector of the node or random and drift is 0. To compute the new embedding of existing node, $\mathbf{h}_{struct}$ is the feature vector of the new node, self-propation uses the most recent embedding of the node and drift is based on previous time point.
- *Both nodes in the observed event are new* — $\mathbf{h}_{struct}$ is the feature vector of the feature vector of the other nodes, $\mathbf{z}(t_0^u)$ s the feature vector of the node or random and drift is 0.

Finally, Algorithm 1 does not require to handle new nodes any differently. As already available in the paper, both $\mathbf{A}$ and $\mathcal{S}$ are qualified by time and hence the matrices get updated every time. The starting dimension of the two matrices can be specified in two ways: (i) Construct both matrices of dimension = total possible no. of nodes in dataset and make the rows belonging to unseen nodes 0. (ii) Expand the dimensions of matrices as you start seeing new nodes. While we implement the first case, (ii) will be required in real-world streaming scenario.

# C    ABLATION STUDY

DyRep framework unifies several components that contribute to its effectiveness in learning rich node representation over complex and nonlinear processes in dynamic graphs. In this section, we provide insights on each component and how it is indispensable to the learning mechanism by performing an ablation study on various design choices of our model. Specifically, DyRep can be divided into three main parts: **Multi-time scale point process model**, **Representation Update Formulation** and **Conditional Intensity Based Attention Mechanism**. We focus on design choices available in each component and evaluate them on large github dataset. DyRep in the Figure 8 is the full model.

**Multiple Time- Scale Processes.** For this component, we perform two major tests:

- **DyRep-Comm.** In this variant, we make Eq 1., time-scale independent (i.e. remove $k$) and we train on only Communication Events. But we evaluate on both communication and association events. Please note that this is possible as our framework can compute representations for unseen nodes. Hence during training they will only learn representation parameters based on communication events. It is observed that compared to the full model, the performance of model degrades in prediction for both types of events. But the decline is more prominent for the Association events compared to Communication Events.

- **DyRep-Assoc.** In this variant, similar to above, we make Eq 1., time-scale independent and we train on only Association Events. But we evaluate on both communication and association events. It is observed that compared to the full model, the performance of model degrades in prediction for both types of events. But the decline is more prominent for the Communication events compared to Association Events.

The above two experiments show that considering events at a single time scale and not distinguishing between the processes hurt the performance. Although the performance is hurt more when communication events are not considered which may be due to the more availability of communication events due to its rapid frequency. We also performed a small test by training on all events but using a single scale parameter ($\psi$). The performance for both the dynamics degrades which demonstrates the effectiveness of $\psi_k$.

**Representation Update Formulation.** For this component, we focus on Eq. 4 and switch off the components to observe its effect.

- **DyRep-No-SP.** In this variant, we switch off the self-propagation component and we observe that the overall performance is not hurt significantly by not using self-propagation. In general, this term provides a very weak feature and mainly captures the recurrent evolution of one's own latent features independent of others. It is observed that the deviation has increased for Association events which may point to the reason that there are few nodes who have links but highly varying frequency of communication and hence most of their features are either self-propagated or completely associated with others.

- **DyRep-No-Struct.** In this variant, we remove the structural part of the model and as one would expect, the performance drops drastically in both the scenarios. This provides evidence to the necessity of building sophisticated structural encoders for dynamic graphs.

**Intensity Attention Mechanism.** For this component, we focus on Section 3.2 which builds the novel intensity based attention mechanism. Specifically, we carry following test:

- **DyRep-No-Att.** Here we completely remove the attention from the structural component and we see a significant drop in the performance.

- **DyRep-S-Comm.** In this variant, we focus on Algorithm 1 and we only make update to the $\mathcal{S}$ matrix for Communication events but do not do it for Association events. This leads to slightly worse performance which helps to see how the $\mathcal{S}$ matrix is helping to mediate the two processes and not considering association events leads to loss of information.

- **DyRep-S-Assoc.** In this variant, we focus on Algorithm 1 and we only make update to the $\mathcal{S}$ matrix for Association events but do not do it for Communication events. This leads to a significant drop in performance again validating the need for using both processes but its prominent effect also shows that communication events (dynamics on the network) is

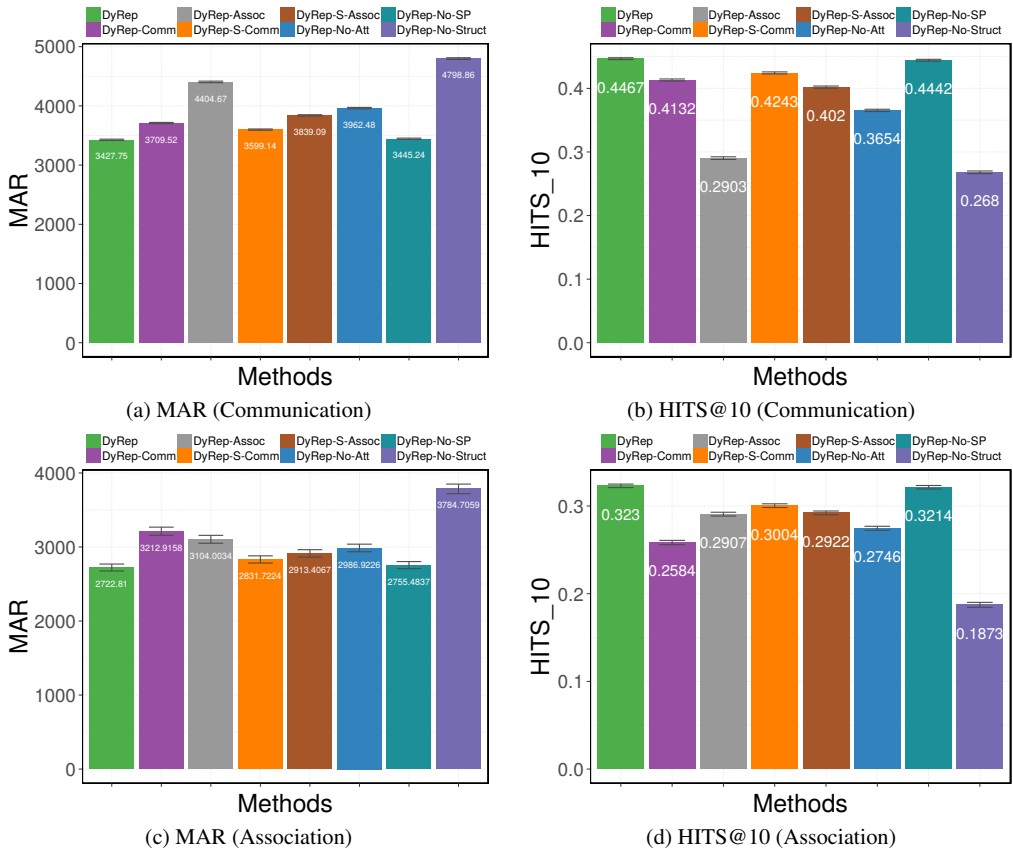

Figure 8: Ablation Study on Github Dataset

## D EXPLORATORY ANALYSIS

We assess the quality of learned embeddings and the ability of model to capture both temporal and structural information. Let $t_0$ be the time point when train ended. Let $t_1$ be the timepoint when the first test slot ends.

**Effect of Association and Communication on Embeddings.** We conducted this experiment on Social dataset. We consider three use cases to demonstrate how the interactions and associations between the nodes changed their representations and visualize them to realize the effect.

- **Nodes that did not have association before test but got linked during first test slot.** Nodes 46 and 76 got associated in test between test points 0 and 1. This reduced the cosine distance in both models but DyRep shows prominent effect of this association which should be the case. DyRep reduces the cosine distance from 1.231 to 0.005. Also, DyRep embeddings for these two points belong to different clusters initially but later converge to same cluster. In GraphSage, the cosine distance reduces from 1.011 to 0.199 and the embeddings still remain in original clusters. Figure 9 shows the visualization of embeddings at the two time points in both the methods. This demonstrates that our embeddings can capture association events effectively.

- **Nodes that did not have association but many communication events (114000).** Nodes 27 and 70 is such a use case. DyRep embeddings consider the nodes to be in top 5 nearest neighbor of each other, in the same cluster and cosine distance of 0.005 which is aligned with the fact that nodes with large number of events tend to develop similar features over time.

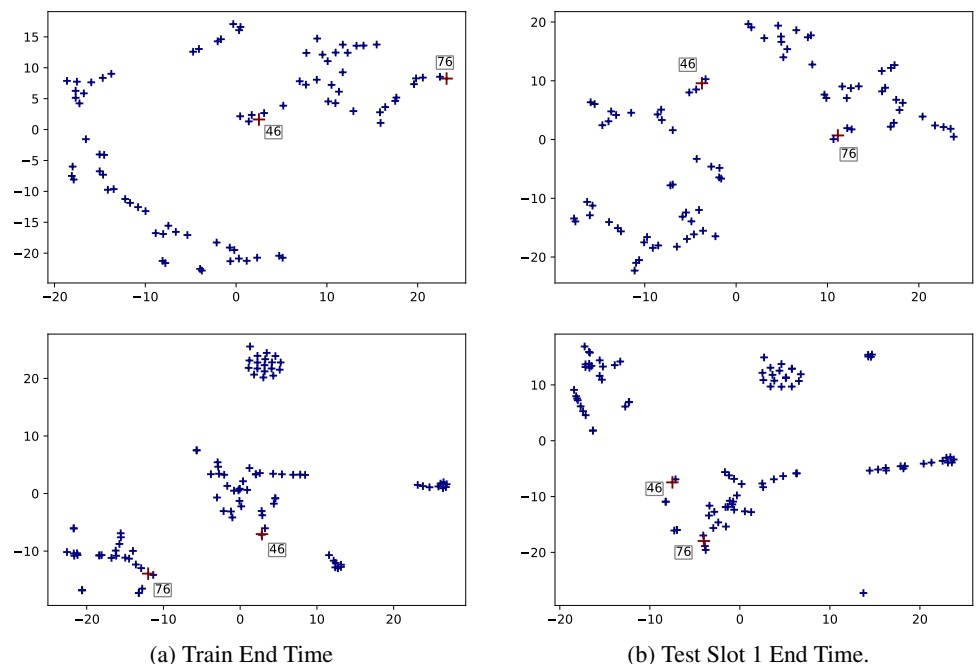

(a) Train End Time

(b) Test Slot 1 End Time.

Figure 9: Use Case I. **Top row:** GraphSage Embeddings. **Bottom Row:** DyRep Embeddings.

Graphsage on the other hand considers them 32nd nearest neighbor, puts them in different clusters with cosine distance - 0.792. Figure 10 shows the visualization of embeddings at the two time points in both the methods. This demonstrates the ability of DyRep's embedding to capture communication events and their temporal effect on embeddings effectively.

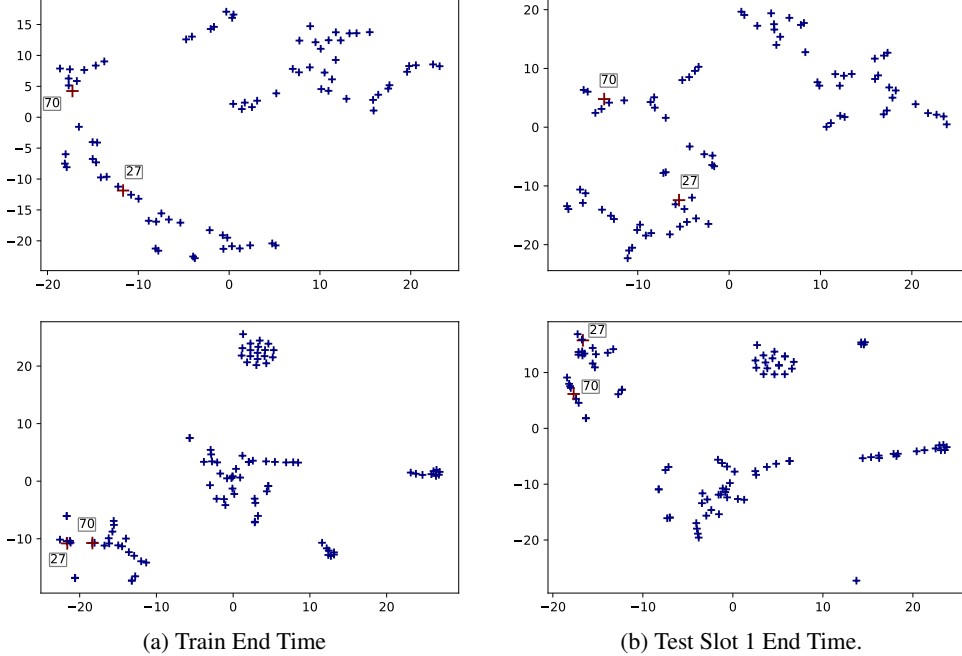

(a) Train End Time

(b) Test Slot 1 End Time.

Figure 10: Use Case II. **Top row:** GraphSage Embeddings. **Bottom Row:** DyRep Embeddings.

- **Temporal evolution of DyRep embeddings.** In figure 11 we visualize the embedding positions of the nodes (tracked in red) as they evolve through time and forms and breaks from clusters.

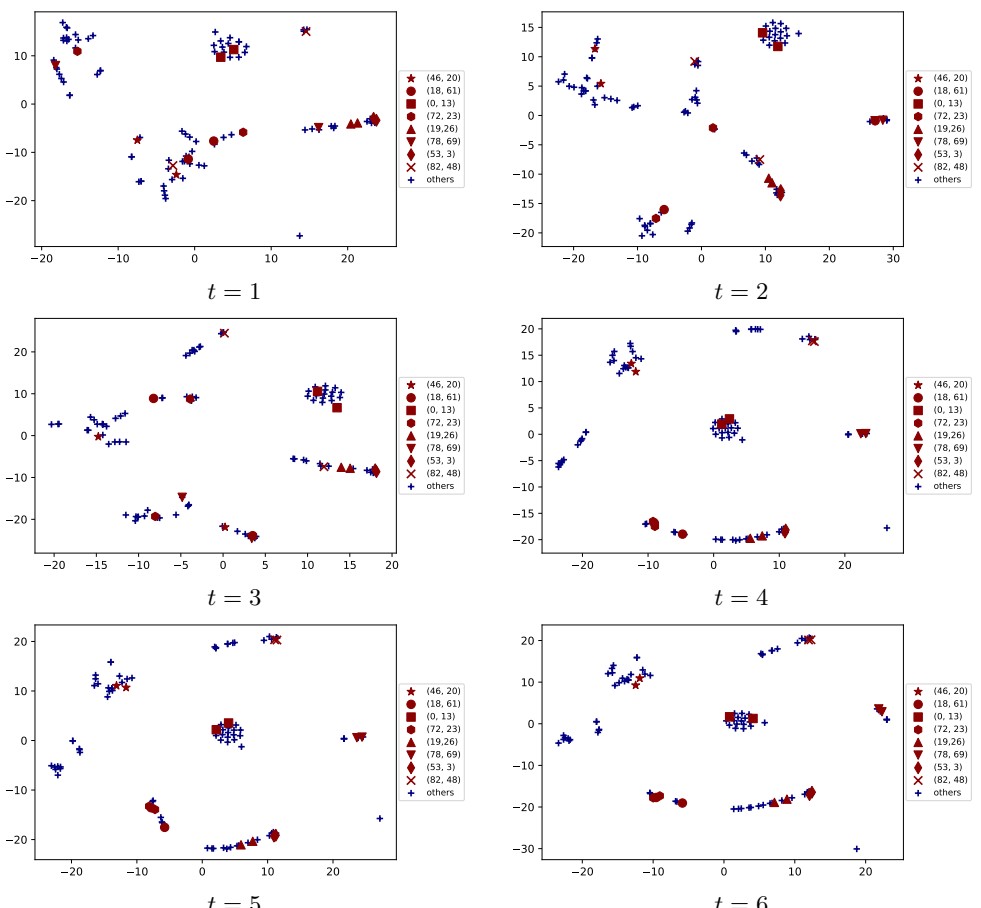

Figure 11: Use Case IV: DyRep Embeddings over time - From left to right and top to bottom. $t$ are the timepoints when test with that id ended. Hence, $t = 1$ means the time when test slot 1 finished.

## E    FULL EXPERIMENT RESULTS FOR BOTH DATASETS

Figure 12 provides HITS@10 results in addition to the MAR results reported for Link Prediction in Section 5 (Experiments) of the main paper.

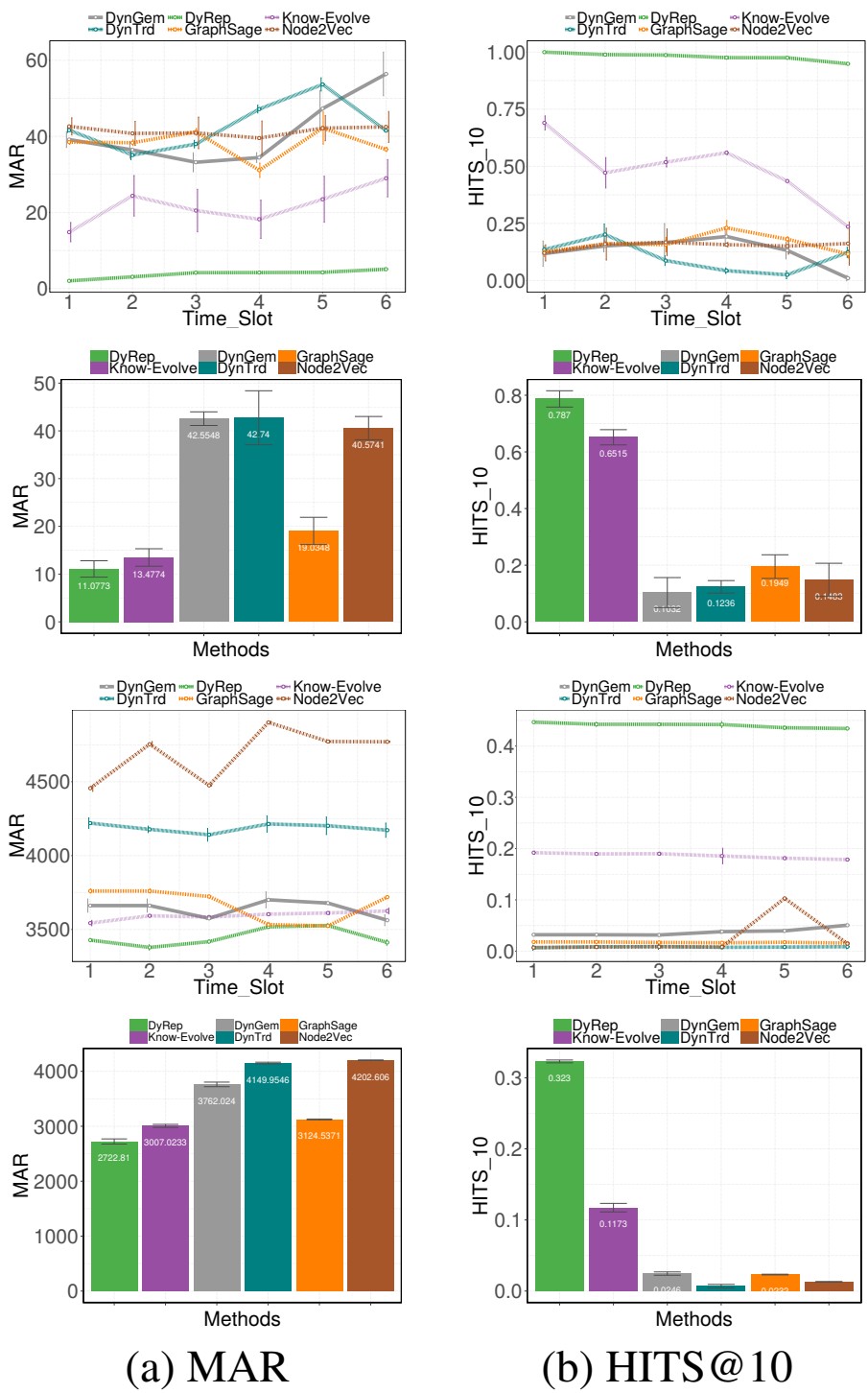

(a) MAR    (b) HITS@10

Figure 12: Dynamic Link Prediction Performance: **Top 2 rows** show performance for Social Evolution Dataset. **Bottom 2 rows** show performance for Github Dataset. 1st and 3rd row show performance for Communication Events while 2nd and 4th row show performance for Association Events.

# F  DETAILED RELATED WORK

**Static Embedding Approaches.** Representation Learning approaches for static graphs can be broadly classified into two categories – Node embedding approaches aim to encode structural information pertaining to a node to produce its low-dimensional representation (Cao et al., 2015; Grover & Leskovec, 2016; Perozzi et al., 2014; Tang et al., 2015; Wang et al., 2016a; 2017; Xu et al., 2017). As they learn each individual node's representation, they are inherently transductive. Recently, (Hamilton et al., 2017a) proposed GraphSage, an inductive method for learning functions to compute node representations that can be generalized to unseen nodes. Sub-graph embedding techniques learn to encode higher order graph structures into low dimensional vector representations (Scarselli et al., 2009; Li et al., 2016; Dai et al., 2016). Further, various approaches to use convolutional neural networks (Kipf & Welling, 2017; 2016; Bruna et al., 2014) over graphs have been proposed to capture sophisticated feature information but are generally less scalable. Most of these approaches only work with static graphs or can model evolving graphs without temporal information.

**Dynamic Embedding Approaches.** Preliminary approaches in dynamic representation learning have considered discrete time approach. (Zhu et al., 2016) propose a temporal latent space model for link prediction using nonnegative matrix factorization. (Goyal et al., 2017) uses a warm start method to train across snapshots and employs a heuristic approach to learn stable embeddings over time but do not model time. (Zhou et al., 2018) focuses on specific structure of triad to model how close triads are formed from open triads in dynamic networks. (Seo et al., 2016) proposes a deep architecture based on combination of CNN to capture spatial characteristics and an RNN to capture temporal characteristics, to model structured sequences which in graph case will lead to discrete time model. (Du et al., 2015) develops extends skip-gram based approaches for network embedding to dynamic setting where the graphs a re observed as discrete time snapshot and the goal is to learn embeddings that can preserve the optimality of skip-gram objective. NetWalk (Yu et al., 2018) is a discrete-time dynamic embedding approach specifically designed for anomaly detection which uses clique based embedding techniques to learn vertex representations. Recently, (Trivedi et al., 2017) proposed Know-Evolve, a deep recurrent architecture to model multi-relational timestamped edges that addresses the communication process. Unlike our approach, Know-Evolve models all edges at a single timescale, works for setting restricted to relational graphs and uses only edge-level structural information with no attention mechanism. DANE (Li et al., 2017) proposes a network embedding method in dynamic environment but their dynamics consists of change in node's attributes over time and their current work can be considered orthogonal to our approach. (Zuo et al., 2018) proposes a dynamic network formation model to learn node representations by employing a Hawkes process to model the temporal evolution of neighborhood for nodes. This work only considers association events. (Ngyuyen et al., 2018) proposes a continuous time embedding framework that employs a temporal version of traditional random walks in a simple manner to capture temporally evolving neighborhood information.

**Other models for dynamic networks.** There exists a rich body of literature on temporal modeling of dynamic networks (Kim et al., 2017) that focus on link prediction tasks but their goal is orthogonal to us as they build task specific methods and do not focus on representation learning. Further, there are several approaches in graph mining and temporal relational learning community (Loglisci & Malerba, 2017; Loglisci et al., 2015; Esteban et al., 2016; Jiang et al., 2016) that consider dynamic networks but are orthogonal to our current work. Research on learning dynamic embeddings has also progressed in linguistic community where the aim is to learn temporally evolving word embeddings (Bamler & Mandt, 2017; Rudolph & Blei, 2018). (Yang et al., 2017; Sarkar et al., 2007) include some other approaches that propose model of learning dynamic embeddings in graph data but none of these models consider time at finer level and do not capture both topological evolution and interactions. (Meng et al., 2018) proposes subgraph pattern neural networks that focuses on evolution of subgraphs instead of single nodes and links. They build a novel neural network architecture for supervised learning where the hidden layers represent the subgraph patterns observed in the data and output layer is used to perform prediction. (Yuan et al., 2017) induces a dynamic graph from videos based on the visual correlation of object proposal that spans across the video. They further propose an LSTM based architecture to capture temporal dependencies over this induced graph and perform object detection. (Jerfel et al., 2017) proposes a dynamic probabilistic model in bipartite case of user-item recommendation where the goal is to learn the evolution of user and item latent features

under the context of Poisson factorization, thus considering the evolution processes of users' and items' latent features as independent of each other.

**Deep Temporal Point Process Models.** Recently, (Du et al., 2016) has shown that fixed parametric form of point processes lead into the model misspecification issues ultimately affecting performance on real world datasets. (Du et al., 2016) therefore propose a data driven alternative to instead learn the conditional intensity function from observed events and thereby increase its flexibility. Following that work, there have been increased attraction in topic of learning conditional intensity function using deep learning(Mei & Eisner, 2017) and also intensity free approach using GANS (Xiao et al., 2017) for learning with deep generative temporal point process models.

## G IMPLEMENTATION DETAILS

### G.1 ADDITIONAL DATASET DETAILS

Table 2: Dataset Statistics for Social Evolution and Github.

| Dataset | #Nodes | #Initial Associations | #Final Associations | #Communications | Clustering Coefficient |
|---|---|---|---|---|---|
| Social Evolution | 83 | 376 | 791 | 2016339 | 0.548 |
| Github | 12328 | 70640 | 166565 | 604649 | 0.087 |

For the social evolution dataset, we consider Proximity, Calls and SMS records between users as communication events (k=1) and all Close Friendship records as association events (k=0). For Github dataset, we consider Star/Watch records as communication events (k=1) and Follow records as association events (k=0). The Social Evolution data is collected from Jan 2008 to to June, 30 2009. We consider the association events between user from Jan 2008-Sep 10, 2008 (survey date) to form the initial network and use the rest of data for our experiments. We collected Github data from Jan 2013 - Dec 2013. For the nodes in 2013, we consider Follow link that existed between them before 2013 to form the initial network. We pre-process both datasets to remove duplicate (not recurrent in time) records and self-loops. We also process Github dataset to only contain users (and not organizations) as nodes and we select nodes that have at least 40 communication (watch) events and 10 association (follow) events.

**Temporal Train/Test Split**: For all the experiments, the data is divided into train and test based on time line. For Social Evolution Dataset, we train on data from Sep 11, 2008 to Apr 30, 2009 and use May 1, 2009-Jun, 30 2009 data for test which gives 10 days of time per test slot. This leads to an approximate 70/30 (train/test) split. For Github data, we train from Jan 1, 2013 to Sep 30, 2013 and test for Oct 1, 2013 - Dec, 31 2013 which gives 15 days of events per time slot. This leads to an approximate 65/35 (train/test) split.

### G.2 TRAINING CONFIGURATIONS

We performed hyper parameter search for best performance for our method and all the baselines and used the following hyper-parameters to obtain the reported results:
– For social dataset: Num nodes = 100, Num Dynamics = 2, bptt (sequence length) = 200, embed_size = 32, hidden_unit_size = 32, nsamples (for survival) = 5, gradient_clip = 100 and no dropout.
– For github dataset: Num nodes = 12328, Num Dynamics = 2, bptt (sequence length) = 300, embed_size = 256, hidden_unit_size = 256, nsamples (for survival) = 5, gradient_clip = 100.

For baselines, we used the implementations provided by their authors and we report the range of configurations used for baseline here: $max\_iter = \{1000, 5000, 10000\}, bptt = \{100, 200, 300\}, lr = \{0.0005, 0.0050.5, 0.1, 1\}, embed\_E = \{32, 64, 128, 256\}, embed\_R = \{32, 64, 128, 256\}, hidden = \{32, 64, 128, 256\}, warm = 0, t\_scale = 0.0001, w\_scale = 0.1, num\_epochs = \{10, 50, 100, 500, 1000\}$. As mentioned in experiment section, we always train baselines with warmstart in a sliding window training fashion.

**Know-Evolve:** The code provided by the authors was implemented in C++.
**GraphSage:** The code was implemented in Tensorflow by the authors. We use only the unsupervised train module to generate embeddings.
**Node2Vec:** We use the original python code with few changes in the hyper-parameters. We fix q in

the node2vec as 0.8 for Social Dataset and 1 for Github dataset.
**DynGEM:** We experiment on the original code implemented in Keras with Theano backend by the authors.
**DynTrd:** We use original code provided by the authors.

For tSNE embedding visualization in Figure 4, we used $sklearn.manifold.TSNE$ library to plot this figure with n components = 2, learning rate = 200, perplexity = 30, metric = "euclidean", min_grad_norm = 1e-9, early exaggeration = 4 and ran for 40,000 iterations.

# H    MONTE CARLO ESTIMATION FOR SURVIVAL TERM IN $\mathcal{L}$ FOR SECTION 4

---

**Algorithm 2** Computation of integral term in $\mathcal{L}$ for a mini-batch

---

**Input:** Minibatch $\mathcal{M} = \{m_q = (u, v, t, k)_q\}_{q=1}^{|\mathcal{M}|}$. Minibatch node list $\mathbf{l}$, sample size $N$.
**Output:** Minibatch survival loss $L_{surv}$

$L_{surv} = 0.0$
**for** $q = 0 \ to \ |\mathcal{M}| - 1$ **do**
   $t_{curr} = m_q \to t; u_{curr} = m_q \to u$
   $v_{curr} = m_q \to v \ ; \ u_{surv} = 0 \ ; \ v_{surv} = 0$
   **for** $N$ samples **do**
      select $u_{other} \in \mathbf{l}$ uniformly randomly s.t. $u_{other} \notin \{u_{curr}, v_{curr}\}$
      select $v_{other} \in \mathbf{l}$ uniformly randomly s.t. $v_{other} \notin \{u_{curr}, v_{curr}\}$
      **for** $k \in \{0, 1\}$ **do**
         $u_{surv}+ = \lambda_k^{u_{curr}, v_{other}}(t_{curr})$
         $v_{surv}+ = \lambda_k^{u_{other}, v_{curr}}(t_{curr})$
      **end for**
   **end for**
   $L_{surv}+ = (u_{surv} + v_{surv})/N$
**end for**
**return** $L_{surv}$

---

Algorithm 2 is a simple variant of Monte Carlo trick to compute the survival term of log-likelihood equation. Specifically, in each mini-batch, we sample non-events instead of considering all pairs of non-events (which can be millions). Let $m$ be the mini-batch size and $N$ be the number of samples. The complexity of Algorithm 2 will then be $\mathcal{O}(2mkN)$ for the batch where the factor of 2 accounts for the update happening for two nodes per event which demonstrates linear scalability in number of events.

