# OpenReview forum: "DyRep: Learning Representations over Dynamic Graphs"
_ICLR.cc/2019/Conference_

### Official Review · AnonReviewer3 · 2018-11-03
**This paper presents a dynamic graph embedding method, which considers two types of dynamics in evolving networks: association events with node and edge grows, and communication events with node-node interactions.**

**Rating:** 8
**Confidence:** 4

**Review:**

The paper is very well written. The proposed approach is appropriate on modeling the node representations when the two types of events happen in the dynamic networks. Authors also clearly discussed the relevance and difference to related work. Experimental results show that the presented method outperforms the other baselines.
Overall, it is a high-quality paper.
There are only some minor comments for improving the paper:
ν	Page 6, there is a typo. “for node v by employing …”  should be “for node u”
ν	Page 6, “Both GAT and GaAN has”   should be  “Both GAT and GaAN have”
ν	In section 5.1, it will be great if authors can explain more what are the “association events” and “communication events” with more details in these two evaluation datasets.

---

> ### Author Response · Authors · 2018-11-14
> **Response to Reviewer 3**
>
> Thank you for your review! We appreciate your time and supportive feedback and we are glad that you find our work interesting. Details about the corresponding association and communication events in the two datasets are provided in Appendix E.1. We uploaded a revised version that contains your suggested changes.

---

### Official Review · AnonReviewer1 · 2018-11-03
**Marked Point Process extension of (Trivedi et al., 2017)**

**Rating:** 7
**Confidence:** 5

**Review:**

Overall, the contribution of the paper is somewhat limited [but a little more than my initial assessment, thanks to the rebuttal]. It is essentially an extension of (Trivedi et al. 2017), adding attention to provide self-exciting rates, applied to two types of edges (communication edges and “friendship” edges). Conditioned on past edges, future edges are assumed independent, which makes the math trivial. The work would be better described as modeling a Marked Point Process with marks k \in {0,1}.
Other comments:
1.	[addressed] DyRep-No-SP is as good as the proposed approach, maybe because the graph is assumed undirected and the embedding of u can be described by its neighbors (author rebuttal describes as Localized Propagation), as the neighbors themselves use the embedding of u for their own embedding (which means that self-propagation is never "really off"). Highly active nodes have a disproportional effect in the embedding, resulting in the better separated embeddings of Figure 4. [after rebuttal: what is the effect of node activity on the embeddings?]
2.	[unresolved, comment still misundertood] The Exogenous Drive W_t(t_p – t_{p−1}) should be more personalized. Some nodes are intrinsically more active than others. [after rebuttal: answer "$W_t(t_p - t_{p-1})$ is personalized as $t_p$ is node specific", I meant personalized as in Exogenous Drive of people like Alice or Bob]
3.	[unresolved] Fig 4 embeddings should be compared against (Trivedi et al. 2017) [after rebuttal: author revision does not make qualitative comparison against Trivedi et al. (2017)]

Besides the limited innovation, the writing needs work.
4.	[resolved] Equation 1 defines $g_k(\bar{t})$ but does not define \bar{t}. Knowing (Trivedi et al. 2017), I immediately knew what it was, but this is not standard notation and should be defined.
5.	[resolved] $g_k$ must be a function of u and v
6.	[resolved] “$k$ represent the dynamic process” = >  “$k$ represent the type of edge” . The way it is written $k$ would need to be a stochastic process (it is just a mark, k \in {0,1})
7.	[resolved] Algorithm 1 is impossibly confusing. I read it 8 times and I still cannot tell what it is supposed to do. It contains recursive definitions like $z_i = b + \lambda_k^{ji}(t)$, where $\lambda_k^{ji}(t)$ itself is a function of $z_i(t)$. Maybe the z_i(t) and z_i are different variables with the same name?
8.	[resolved] The only hint that the graph under consideration is undirected comes from Algorithm 1, A_{uv}(t) = A_{vu}(t) = 1. It is *very* important information for the reader.
Related work (to be added to literature):
Dynamic graph embedding: (Yuan et al., 2017) (Ghassen et al., 2017)
Dynamic sub-graph embedding: (Meng et al., 2018)

Minor:
state-of-arts => state-of-the-art methods
list enumeration “1.)” , “2.)” is strange. Decide either 1) , 2) or 1. , 2. . I have never seen both.
MAE => mean absolute error (MAE)

Yuan, Y., Liang, X., Wang, X., Yeung, D. Y., & Gupta, A., Temporal Dynamic Graph LSTM for Action-Driven Video Object Detection. ICCV, 2017.
Jerfel,  , Mehmet E. Basbug, and Barbara E. Engelhardt. "Dynamic Collaborative Filtering with Compound Poisson Factorization." AISTATS 2017.
Meng, C., Mouli, S.C., Ribeiro, B. and Neville, J., Subgraph Pattern Neural Networks for High-Order Graph Evolution Prediction. AAAI 2018.

--- --- After rebuttal

Authors addressed most of my concerns. The paper has merit and would be of interest to the community. I am increasing my score.

---

> ### Author Response · Authors · 2018-11-14
> **Response to Reviewer 1 - Part I**
>
> Thank you for your review! We appreciate your comments and suggestions.
>
> As a preface to our response, we wish to mention that, unlike existing approaches, our  work expresses dynamic graphs at multiple time-scales as follows:
> a)  Dynamic ”of” the Network:  This corresponds to the topological changes of the network – insertion or deletion of nodes and edges. We use "Association" to label the observed process corresponding to this dynamic.
> b)  Dynamic ”on” the Network:  This corresponds to  activities on a *fixed* network topology – self evolution of node’s features,  change in node’s features due to exogenous drive (activities external to network),  information  propagation  within  network  and  interactions  between nodes which may or may not have direct edge between them. We use "Communication" to label the observed process of interaction between nodes (only the observed part of dynamic ”on” the network).
>
> General Comment:
> ==============
> Overall, the contribution of the paper is limited. It is essentially a minor extension of (Trivedi et al. 2017), adding attention, applied to two types of edges (communication edges and “friendship” edges). Edges are assumed independent, which makes the math trivial. The work would be better described as modeling a Marked Poisson Process with marks k \in {0,1}.
>
> Response:
> =========
> We politely disagree with these comments as this is an incorrect characterization of our work. It seems that the misunderstanding arises from your assumption (including point 6) that ‘k’ is type of an edge, ‘k’ is a mark and ‘k’ has independence, none of which is true. ‘k’ truly distinguishes scale of event dynamics (not type of edge) in our two-time scale model. In fact, when k=1, it is an interaction event which is not considered as an edge between nodes in our model. The edge (which forms graph structure) only appears through an association event (k=0). Indeed, ‘k’ corresponds to stochastic processes at different time scales and hence $\psi_k$ is the rate (scale) parameter corresponding to each dynamic. Further, every time when k=0, an edge is created between different node pairs. As we clearly mention in the paper, we do not consider edge type in this work and hence ‘k’ is not a mark. However, edge type can be added to Eq 4 in case it is available. Finally, dynamic processes realized by k=0 and k=1 are not independent and are highly interleaved in a nonlinear fashion. For instance, formation of a structural edge (k=0) affects interactions (k=1) and vice versa. Algorithm 1 captures this intricate dependencies as we will describe below. Based on the above points, it follows that our model is not a marked Poisson process. In fact, it does not take any specific form of point process - rather learns the conditional intensity function through a function approximation.
>
> In terms of contributions, we argue that our approach of modeling dynamic graphs at multiple scales and learning dynamic representations as latent mediation process bridging the two dynamic processes, is a significant innovation compared to any existing approaches. This is a non-trivial effort for a setting where the dynamic processes evolve in a complex and nonlinear fashion. Further, our temporal point process based structural-temporal self-attention mechanism to model attention based on event history of a node is very novel and has not been attempted before. Our attention model can: 1) take into account temporal dynamics of activities on edge and 2) capture effects from faraway nodes due to dependence on event history. This is a formal advancement to state-of-the-art models of non-uniform attention (such as Graph Attention networks).
>
> Further, the paper provides an in-depth comparison with (Trivedi et. al. 2017) (including Table 1). Here we reiterate the differences: (Trivedi et. al. 2017) model events at single time scale and do not distinguish between two dynamic processes. They only consider edge level information for learning the embeddings. Our model considers a higher order neighborhood structure to compute embeddings. More importantly, in their work, the embedding update  for a node ‘u’ considers the edge information for the same node ‘u’ at a previous time step. This is entirely different from our structural model based on ”Localized Embedding Propagation” principle which states: Two nodes involved in an event form a temporary (communication) or a permanent (association) pathway for the information to propagate from the neighborhood of one node to the other node. This means, during the update of embedding for node ‘u’, information is propagated from the neighborhood of node ‘v’ (and not node ‘u’, please check Eq. 4) to node ‘u’. Subsequently, (Trivedi et. al. 2017) does not have any attention mechanism as they don't consider structure.

---

> > ### Comment · AnonReviewer1 · 2018-11-19
> > **I also politely disagree, but mostly because some of my comments were misunderstood**
> >
> > Thank you for your reply. I do realize now that the process is not Poisson as the definition of \lambda clearly depends on past marks (it is not an externally driven process like a non-homogeneous Poisson process). I will change my review accordingly.
> >
> > I also apologize but I fear we are talking past each other here (“We disagree with these comments as this is an incorrect characterization of our work” … ). I will strive to be more specific from now on.
> >
> > “It seems that the misunderstanding arises from your assumption (including point 6) that … ‘k’ is a mark” => By your own definition of O = \{(u, v, t, l, k)_p\}_{p=1}^P , which fits the Definition 2.1.2 of Jacobsen (2006) where T_p is your p-th event time and Y_p = (u, v, l, k) is an element of a Polish space E. When you say O is a not a Marked point process, what is the basis for the claim? Why would Y_p not be represented by a Polish space?
> >
> > Formally, any time-varying graph is a Marked point process where the edges are the marks. When I say “Graph process”, it is implicit that it has edge marks. Thus, my comment “Graph process” with edge marks k implies a measure (density) over the sigma algebra (sequence) given by O = \{(u, v, t, k)_p\}_{p=1}^P. The variable “l” is not properly a mark because it can be re-constructed from the process (l_p = 1 if there has been any event with k=0 in the past). Algorithm 1 uses this marks definition when it does “if k = 0 then Auv(t) = Avu(t) = 1“, i.e., k=0 is a mark of an observable edge (see description next).
> >
> > “It seems that the misunderstanding arises from your assumption (including point 6) that ‘k’ is type of an edge,
> > Possibly my general use of the ill-defined term “edge” was not clear. I am thinking of (u,v) as a tuple. If (u,v) is a physical edge or a virtual edge “interaction”, k \in \{0,1\} defines a mark (physical or virtual).
> >
> > “It seems that the misunderstanding arises from your assumption (including point 6) that ‘k’ has independence, none of which is true.”
> > We seem be to talking about different things. Marks (u, v, t, k) are conditionally independent given the model and past marks, per your likelihood \mathcal{L}. This is the independence I was referring to. Adding these marks to Trivedi et al. (2017) is rather (mathematically) straightforward given the independent nature of the model. Mathematically straightforward does not mean it is easy to get it to work in practice and releasing the code would be important.
> >
> > Jacobsen, Martin. Point process theory and applications: marked point and piecewise deterministic processes. Springer Science & Business Media, 2006.
> >
> > Minor:
> > Page 3, λ(t)dt:= P[event .. ] missing brackets

---

> > > ### Author Response · Authors · 2018-11-23
> > > **We agree and provide further clarifications**
> > >
> > > Thank you for a detailed response. We believe that we were describing similar things but from different perspectives and your response has greatly helped us to distill that. Below we provide further clarifications on our perspective:
> > >
> > > First, we clarify that $l$ was only used for book-keeping to check the status of link in Algorithm 1, so it should not be part of event representation $e$ and we rectify that in our revision by removing it completely as adjacency matrix A already provides that information.
> > >
> > > Marked Process: From a mathematical viewpoint, we agree with you that for any event $e$ at time $t$, any information other than the time point can be considered a part of mark space describing the events. Hence, in our case, given a one-dimensional timeline, we can consider O=\{(u,v,k)_p, t_p)_{p=1}^P as a marked process with the triple (u,v,k) representing the mark.
> > >
> > > However, using a single-dimensional process with such marks does not allow to efficiently and effectively discover or model the structure in the point process useful for learning intricate dependencies between events, participants of the events and dynamics governing those events. Hence, it is often important to extract the information out of the mark space and build an abstraction that helps to *discover the structure* in point process and make this learning *parameter efficient*. In our case, this translates to two components:
> > >
> > > i) The nodes in the graph are considered as dimensions of the point process, thus making it a multi-dimensional point process where an event represents interaction/structure between the dimensions, thus allowing us to explicitly capture dependencies between nodes.
> > > ii) The topological evolution of networks happen at much different temporal scale than activities on a fixed topology network (e.g. rate of making friends vs liking a post on a social network). However both these processes affect each other’s evolution in a complex and nonlinear fashion.  Abstracting $k$ to associate it with these different scales of evolution facilitates to model our purpose of expressing dynamic graphs at two time scales in a principled manner. It also provides an ability to explicitly capture the influential dynamics (Chazelle et. al. 2012) of topological evolution on dynamics of network activities and vice versa (through the  learned embedding -- aka evolution through mediation which is the most crucial part of this whole framework).
> > >
> > > Note that this distinction in use of mark information is also important as we learn representations for nodes (dimensions) but not for $k$. Our overall intention here is to make sure that $k$ representing two different scales of event dynamics is not confused with edge or interaction type. For instance, in case of typed structural edge (e.g. wasbornIn, livesIn) or typed interaction (e.g. visit, fight etc. as in Trivedi et. al. 2017), one would add type as another component in the mark space to represent an event while $k$ still signifying different dynamic scales. In that sense, (Trivedi et. al. 2017) can also be viewed as a marked process that only models the typed interaction dynamics at a single time-scale and does not model topological evolution.
> > >
> > > Independence: We agree with you but we would paraphrase your statement as follows: The next event and its mark (u,v,k) at time $t$ is conditionally independent of all past events and their marks given the conditional intensity function, which itself is a function of the model and the most recent *learned representations* of nodes (this is the most important part for this to hold) at time $t$.
> > >
> > > Bernard Chazelle. Natural Algorithms and Influence Systems, 2012.

---

> ### Author Response · Authors · 2018-11-14
> **Response to Reviewer 1 - Part II**
>
> Responses to Other Comments:
> ========================
>
> 1) This is incorrect as self-propagation mainly captures the recurrent evolution of one’s own latent features independent of others. Self-propagation principle states: A node evolves in the embedded space with respect to its previous position (e.g. set of features) and not in a random fashion. Based on Localized Propagation principle described above, a node's embedding is described by information it receives from other node and not exclusively it's own neighbors. The good performance of DyRep-No-SP signifies that the Localized Propagation term in Eq 4. is able to account for the relative position of node with respect to its previous position more often than not. Further, both dynamic of network and dynamic on network contribute to updates to a node's embedding. The interplay of multi-scale temporal behavior of these processes and evolving features leads to better discriminative embeddings, not just the rate of activities - this is evident by other exploratory use cases we discuss.
>
> 2) $W_t(t_p - t_{p-1})$ is personalized as $t_p$ is node specific.
>
> 3,4) We add the suggested changes to the revised version.
>
> 5) The intention for the *qualitative* exploratory analysis was not to make a performance comparison, which is already available against dynamic baselines in our *quantitative* predictive analysis. The goal of Figure 4 and appendix experiments is to draw the comparison between how embeddings learned using state-of-the-art static methods would differ from our dynamic model in terms of capturing evolving properties over time. To our knowledge, such extensive analysis for dynamic embeddings is not available in previous works. Further, we believe that visualizing embeddings from another dynamic method against our model may not provide informative insights.
>
> 6) This is incorrect - please check our main response above
>
> 7) “z" in Algorithm 1 is a temporary variable whose scope is limited to the algorithm. Please note that $\lambda$ is an input to the algorithm and hence “z" within Algorithm 1 has no interaction with the node embedding z (which always has a superscript) used throughout the paper. Hence, there is no recurrence, however, to avoid any further confusion, we change the temporary variable to “y".
>
> Details explaining Algorithm 1 in full are available on Page 7. Here we provide a simplified high-level explanation. As a starting point, we refer you to the point 2 in paragraph before Eq 4 page 5.  To capture the effect described there, we parameterize the attention module with element of matrix S corresponding to an existing edge that signifies information/effect propagated by that edge. Algorithm 1 computes/updates this S matrix. Please note that S is parameter for a structural temporal attention which means temporal attention is only applied on structural neighborhood of a node. Hence, the value of S are only updated/active in two scenarios: a) the current event is between nodes which already has structural edge (communication between associated nodes or l=1, k=1) and b) the current event is an association event (l=0, k=0). Now, given a neighborhood of node ‘u’, $b$ represents background (base) attention for each edge which is uniform attention based on neighborhood size. Whenever an event occurs between two nodes, this attention changes in following ways: For case (a), just change the attention value for corresponding S entry using the intensity of the event. For case (b), repeat same as (a) but also adjust the background attention for each node as the neighborhood size grows in this case.
>
> 8) Thank you for pointing this. It is true that we consider undirected graphs in proposed work. However, our model can be easily generalized to directed graphs. Specifically, the difference would appear in the update of matrix A used in Algorithm 1, which would subsequently lead to different neighborhood and attention flow for each node. We will add this clarification in the revised paper.
>
> We have uploaded a revised version of the paper to add the above clarifications, address your points and discuss related work cited by you (thank you for the pointers). Please let us know if something is still not clear and we will be happy to further discuss and address your concerns.

---

> ### Author Response · Authors · 2018-11-23
> **Thank you for the update**
>
> Thank you for updating your review. We added a clarification on the point process perspective as a response to your previous comment. Here we address your updated review comments and re-emphasize the contributions of our work:
>
> Exogenous Drive: Do you mean Alice/Bob is a person inside network? The exogenous drive constitutes the changes in features of node caused by external influences. However, activities external to network are not observed in the dataset. Hence for a node $u$ (or Alice which will be a node in social network) , the term allows a smooth latent approximation of change in $u$’s features over time caused by such an external effect. Please note, $\bar{t_{p}}^u$ is not the time of previous event in the global dataset, it is time for previous event of node $u$.
>
> Contributions: While one can augment the event specification in (Trivedi et. al. 2017) with additional mark information, that itself is not adequate to achieve our proposed method of modeling dynamical process over graphs at multiple time scales. A subtle but key difference in our deep point process formulation that allows us to achieve our goal of two time-scale expression,  is the form of conditional intensity function (Eq 3 in our paper). We employ a softplus function for $f$ which contains a dynamic specific scale parameter $\psi_k$ to achieve this while (Trivedi et al. 2017) uses an exponential (exp) function for $f$ with no such parameter. The exponential choice of $f$ also restricts their model to Rayleigh dynamics while DyRep can capture more general dynamics.
>
> However, we wish to emphasize that our major contributions for learning dynamic graph representation in this work extend well beyond this conditional intensity function. To the best of our knowledge, our work is the first to adopt the paradigm of expressing network processes at different time-scales (widely studied in network dynamics literature) to representation learning over dynamic graphs and propose an end-to-end framework for the same. Further our novel representation learning module that incorporates *graph structure* - using  Temporal Point Process based Self-Attention (a principled advancement over all existing graph based neural self-attention techniques) and Localized Embedding Propagation - is not a straightforward extension or variant of (Trivedi et al. 2017).We will release the code and datasets with the final version of the paper.
>
> We again thank you for your time and discussions. Please let us know if there are still unclear points and we would be happy to clarify your further concerns.

---

### Official Review · AnonReviewer4 · 2018-11-14
**An interesting idea which could use clearer theoretical justification and larger scale experimental validation.**

**Rating:** 6
**Confidence:** 4

**Review:**

Overall the paper suffers from a lack of clarity in the presentation, especially in algorithm 1, and does not communicate well why the assumption of different dynamical processes should be important in practice. Experiments show some improvement compared to (Trivedi et al. 2017) but are limited to two datasets and it is unclear to what extend end the proposed method would help for a larger variety of datasets.

Not allowing for deletion of node, and especially edges, is a potential draw-back of the proposed method, but more importantly, in many graph datasets the type of nodes and edges is very important (e.g. a knowledge base graph without edges loses most relevant information) so not considering different types is a big limitation.

Comments on the method (sections 2-4).

About equation (1):
 \bar{t} is not defined and its meaning is not obvious. The rate of event occurrence does not seem to depend on l (links status) whereas is seems to be dependent of l in algorithm 1.

I don’t see how the timings of association and communication processes are related, both \lambda_k seem defined independently. Should we expect some temporal dependence between different types of events here? The authors mention that both point processes are “related through the mediation process and in the embedding space”, a more rigorous definition would be helpful here.

The authors claim to learn functions to compute node representations, however the representations z^u seem to be direct embeddings of the nodes. If the representations are computed as functions it should be clear what is the input and which functional form is assumed.

I find algorithm 1 unclear and do not understand how it is formally derived, its justification seems rather fuzzy. It is also unclear how algorithm 1 relates to the loss optimisation presented in section 4.

What is the mechanism for addition of new nodes to the graph? I don’t see in algorithm 1 a step where nodes can be added but this might be handled in a different part of the training.

Comments on the experiments section.

Since the proposed method is a variation on (Trivedi et al. 2017), a strong baseline would include experiments performed on the same datasets (or at least one dataset) from that paper.

It is not clear which events are actually observed. I can see how a structural change in the network can be observed but what exactly constitutes a communication event for the datasets presented?

---

> ### Author Response · Authors · 2018-11-23
> **Response to Reviewer 4 - Part 1**
>
> We thank the reviewer for providing detailed comments. Below we provide clarifications on your specific points:
>
> - Importance of Two-time scale Process: We emphasize that the two-time scale expression of dynamic processes over graphs is not an assumption of our work; it is a naturally observed phenomenon in any dynamic network. For instance, consider the dynamics over a social network. The growth of network (topology change) by addition of new users (nodes) or new friendships (edges) occurs at significantly different rate/dynamics compared to various activities on a *fixed* network topology (self evolution of user’s features, effect on user from activities external to network, information propagation on network or interactions (sending a message, liking a post, comments, etc.). Further, both these dynamics affect each other significantly - befriending someone on social network increases the likelihood of activities between those nodes and on the other way around, activities such as regularly liking or sharing a post or mere prolonged interest in posts from friends of friends may lead to a friendship or follow edge between non-friends.
>
> This dichotomy of expressing network processes at two different time-scales (dynamic *of* the network or network evolution) and (dynamic *on* the network or network activities) is a widely known phenomenon that is subject of several studies in dynamic networks literature [1,2,3,4,5]. However, to the best of our knowledge, our work is the first to adopt this paradigm for large scale representation learning over dynamic graphs and propose an end-to-end framework for the same.
>
> - Support for Node and Edge Types is inherent in our approach and not a limitation of our model. As both node and edge types are essentially features, our model does not require any modification in the approach incorporate them. We have added a brief discussion in Appendix B to explain how our model works in presence of them. Consequently, DyRep can learn representations over various categories of dynamic graphs including but not limited to social networks, biological networks, dynamic knowledge graphs etc. as long as data provides time stamped events for both network evolution and activities on the network.
>
> - Support for Deletion: Being a continuous-time model, our work captures fine-grained temporal dependencies among network processes. To achieve this, the model needs time stamped edges for graphs. However, as we mention in conclusion of our paper, it is difficult to procure data with fine grained deletion time stamps. Further, the temporal point process model requires more sophistication to support deletion. For example, one can augment the model with a survival process formulation to account for lack of node/edge at future time which is an involved task and requires a dedicated investigation outside the scope of this paper.
>
> - Temporal Dependence between events: $lambda$ is the conditional intensity function the *conditional* part represents the occurrence of current event conditional on all past events. Hence, $\lambda(t)$ can also be written as $\lambda(t|\amthcal{H}_t)$ to mention the conditional part where $\mathcal{H}_t$ represents history of all previous event occurrences. In the point process literature,  $\mathcal{H}_t$ is often omitted as it is well understood. Next, the conditional intensity function is derived based on the most recent embeddings of the two nodes in the event. However the node embeddings get updated after every event (whether k = 0 or k=1). For instance, consider that a node $u$ was involved in a communication event (k=1) at time $t1$,  association event (k=0) at time $t2$ and another communication event (k=1) at time $t3$ ($t1$ < $t2$ < $t3$). In this case, the conditional intensity function computed for time $t3$ (when k = 1) will use most recent embeddings of node $u$ updated after its event at time $t2$ (when k =0) and similarly  the conditional intensity function computed for time $t2$ (when k=0) will use most recent embeddings of node $u$ updated after its event at time $t1$ (when k=1). This is how the two processes are interleaved with each other through evolving representations whose learning is the latent mediation process.
>
> [1] Bernard Chazelle. Natural Algorithms and Influence Systems, 2012.
> [2] Damien Farine. The dynamics of transmission and the dynamics of networks, 2017.
> [3] Oriol Artime et. al., Dynamics on networks:  competition of temporal and topological correlations, 2017.
> [4] Haijun Zhou et. al., Dynamic pattern evolution on scale-free networks, 2005.
> [5] Farajtabar et. al., Coevolve:  A Joint Point Process Model for Information Diffusion and Network Evolution, 2015.

---

> ### Author Response · Authors · 2018-11-23
> **Response to Reviewer 4 - Part 2**
>
> - Functional form of Computing Representation: Eq 4. provides the functional form that computes the representations with inputs being the three terms and parameterized by the W parameters. We state this clearly in revised version. Note that z^u(t) in Eq. 4 is qualified by $t$ and it keeps getting updated as the node $u$ gets involved in events. It does not represent direct embedding, rather just the placeholder for evolving embedding. For learning direct embedding of nodes (as done in *transductive* setting), one needs to have node-specific parameter i.e. one dimension of parameter matrix need to be of size = number of nodes in graph. In contrast to that, our setting is *inductive* where the parameters are not node-specific and hence it allows to learn general functions to compute representations given input information for a node. This allows to compute node embeddings for new (unseen) nodes without any necessity of altering the parameter space. This difference in transductive vs. inductive settings is well summarized for graphs in (Hamilton et. al. 2017).
>
> - Algorithm 1: It seems there is a misunderstanding on this point. Algorithm 1 is not a part of training (Algorithm 2 makes training tractable). Algorithm 1 constitutes a vital part of the forward pass (our novel Temporal Point Process based Attention mechanism) that computes node embeddings. As Algorithm 1 is used in an involved process, we believe that a figure accompanying the process may provide easier access to the mathematics behind it. To this end, we have now added an auxiliary figure in Appendix A describing the use of Algorithm 1 and how the whole process works. In addition to the accompanying figure, we have also updated the description of Algorithm 1 in the main paper to make it more readable in the revised version.
>
> - Adding new nodes: It is important to note the *inductive* ability of our framework described in response to your above question on computing functions, as that gives us an inherent ability to support new nodes. In practice, as described in Section 2.3 of the paper, the data contains a set of dyadic events ordered in time. Hence, each event involves two nodes $u$ and $v$. A new node will always appear as a part of such an event and it will be processed by the framework like any other node. We provide some more details on the mechanism in Appendix B.
>
> - Comments on Experiment Section: Both datasets in (Trivedi et. al. 2107) are purely interaction datasets (i.e. contains information about activities on the network, e.g. visit, fight, etc.) but do not consider any topological events i.e. there do not exist an underlying topology between the nodes that interact in those events. One way to remedy that would be to augment such a dataset with an underlying fixed topology knowledge graph such as Freebase or Wikidata. We considered this approach but the issue in this case is the absence of time points for the formation of topological edges. As we require time-stamped events, we chose the datasets that naturally provided both network evolution and activities on network with timestamps in lieu of constructing an artificial network by combining multiple sources where the quality of such construction will also play a role. We believe that the two datasets used in this work contain lot of interesting properties observed in real-world dynamic graphs that helps to adequately evaluate our proposed contributions and serve as a strong empirical evidence of the success of our approach.
>
> In the interest of space, we provide preliminary details on datasets in Section 5.1 while more details on the two datasets are available in Appendix G.1.
>
> Please let us know if something is still not clear and we will be happy to further discuss and address your concerns.
>
> William L. Hamilton et. al., Representation Learning on graphs: Methods and Applications, 2017.

---

### Public Comment · ~Michael_Bronstein1 · 2018-09-30
**prior works on graph deep learning**

I would like to draw the authors' attention to multiple recent works on deep learning on graphs directly related to their work. Among spectral-domain methods, replacing the explicit computation of the Laplacian eigenbasis of the spectral CNNs Bruna et al. with polynomial [1] and rational [2] filter functions is a very popular approach (the method of Kipf&Welling is a particular setting of [1]). On the other hand, there are several spatial-domain methods that generalize the notion of patches on graphs. These methods originate from works on deep learning on manifolds in computer graphics and recently applied to graphs, e.g. the Mixture Model Networks (MoNet) [3] (Note that Graph Attention Networks (GAT) of Veličković et al. are a particular setting of the MoNet [3]). MoNet architecture was generalized in [4] using more general learnable local operators and dynamic graph updates. Finally, the authors may refer to a review paper [5] on non-Euclidean deep learning methods.


1. Convolutional Neural Networks on Graphs with Fast Localized Spectral Filtering, arXiv:1606.09375

2. CayleyNets: Graph convolutional neural networks with complex rational spectral filters, arXiv:1705.07664,

3. Geometric deep learning on graphs and manifolds using mixture model CNNs, CVPR 2017.

4. Dynamic Graph CNN for learning on point clouds, arXiv:1712.00268

5. Geometric deep learning: going beyond Euclidean data, IEEE Signal Processing Magazine, 34(4):18-42, 2017

---

> ### Author Response · Authors · 2018-10-05
> **Thank you for interesting pointers**
>
> We view the work on geometric deep learning as a very interesting direction for representation learning over graphs. However, most current works including cited papers in geometric deep learning over graphs primarily deal with static graphs, while our work focuses on dynamic graphs to jointly model both - topological evolution (dynamic of the network) and node interactions (dynamic on the graph).  It would be interesting complimentary direction to extend cited spectral/spatial domain methods to derive local graph operators that can take into account both both temporal and spatial dynamics. We will add a related discussion section in the updated version of the paper.

---

### Public Comment · (anonymous) · 2018-10-17
**comment**

The paper presents its content in the most complicated way. It defines new concepts of Association (refers to topological evolution) and Communication (refers to node interactions) for dynamic graphs and formulate the problem based on them. In reality, dynamic networks are represented by insertion and deletion of nodes and insertion or deletion of edges between existing nodes. The edges and nodes may have features or labels. The paper defines two new concepts of communication and association which I think are inherited from the edge concept with subtle differences. Association has global effects and communication has local effects on information exchange. I am really confused if we really need to define such new concepts and then propose a model for that, while in reality dynamic graphs usually do not contain these kinds of constraints. Assuming we have the realization of these concepts, can we formulate the problem using simpler models such as networks with typed edges or weighted edges? I am skeptical about how the authors use the datasets in the experiment. For example, in the Social Evolution Dataset, what is association and what is communication? How did you interpret the dataset to find these concepts? Do we really need to consider these concepts in the Social Evolution Dataset to do the link prediction? I think authors can elaborate on new concepts definitions and necessity for considering them in their method.

---

> ### Author Response · Authors · 2018-10-23
> **Response to comment**
>
> Thank you for your interest in our work.
>
> Inspired from [1], our  work expresses dynamic graphs at multiple scales as follows:
> a.)  Dynamic ”of” the Network:  This corresponds to the topological changes in network – insertion or deletion of nodes and edges
> b.)  Dynamic ”on” the Network:  This corresponds to various activities in the network – self evolution of node’s interests/features,  change in node’s features due to exogenous drive (activities external to net-work),  information  propagation  within  network  and  within-network interactions  between nodes which may or may not have direct edge between them.
>
> We  do not  define  "Association"  and  "Communication"  as  two  new  concepts  or constraints  on  dynamic  graphs neither do we claim that in the paper.   Instead,  we  use  those  two  words  to  label  the  well-known  and  naturally *observed* processes corresponding to the dynamics mentioned in (a) and (b) – Association events maps to observed insertion of nodes or edges and Communication events maps to observed interactions between nodes (which is observed part of dynamic ”on” the network).  Nevertheless, this dichotomy of dynamic network processes is well-known and has been subject of several studies [1,  2,  3,  4,  5] in segregated manner.  But none of the existing machine learning approaches has jointly modeled them for representation learning over dynamic graphs (our key objective) to the best of our knowledge.
>
> ”In  reality,  dynamic  networks  are  represented  by  insertion  and  deletion  of  nodes  and  insertion  or  deletion of edges between existing nodes.”
>
> This is a rather limited or constrained view of dynamic graphs as there are many dynamic processes (as listed in b above) occurring on such a graph which cannot be realized by just modeling growth or shrinkage of graph. Approaches based on such model of dynamic network cannot distinguish or model interleaved evolution of network processes which leads to multiple shortcomings:
> – Such a model may capture structural evolution, but it lacks the ability to effectively and correctly capture dynamics ”on” the network.  Concretely, the dynamic process under which a node’s features evolve or node interactions happen within a network (thus leading to information propagation) has vastly different behavior from the dynamic  process  that  leads  to  growth  (shrinkage)  of  the  network  structure.   For  example,  social  network activities such as liking a post or posting on discussion or sharing a video happen at much accelerated rate compared to slow rate of making friends and thereby growing the network.  Hence it is important to express dynamic graphs at different time scales.
> – Edge types only serve as  feature information and they can be readily added in our model if available. Edge weights may or may not be available apriori and may need to be inferred. Both of them are insufficient to effectively model the evolutionary multi-time scale dynamics of structure and network activities and their influence on each other.  Further, neither of them express node specific dynamic properties.  This, in turn, will not help to learn the effect of evolving node representations on observed processes and vice versa.
>
> Extended Details on use of both datasets is available in Appendix E.
>
> [1] Natural algorithms and influence systems.
> [2] The dynamics of transmission and the dynamics of networks.
> [3] Dynamics on networks:  competition of temporal and topological correlations.
> [4] Dynamic pattern evolution on scale-free networks.
> [5] Coevolve:  A Joint Point Process Model for Information Diffusion and Network Evolution.

---

### Author Response · Authors · 2018-11-23
**Revision**

We’d like to thank all the reviewers for your helpful comments. We’ve made the following updates to our paper based on your feedback:

Main paper:
=========
- Revised Section 2.3 based on reviews and discussion.
- Removed $l$ as it was only used for book-keeping and only invoked in Algorithm 1. However, as input to Algorithm 1 is A(\bar{t}), the most recent adjacency matrix, $l$ is redundant and can be removed. This helps to make a cleaner presentation
- Revised the text under Section 3.2.1 including explanation of Algorithm 1 and also rectified minor notations.
- Made \bar notation consistent to signify past time points. Henceforth, for an event at time $t_p$, $\bar{t_p}$ represents the global timepoint just before the current event while for a node $u$ involved in current event at time $t_p$, $\bar{t_p}^u$ represents the timepoint of previous event involving node $u$. This makes all notation consistent and removes any use of $t_{p-1}$ in Eq 4 and $t-$ in Algorithm 1.
- Rectified any minor flaws suggested by the reviewers.

Appendix:
=======
Added two new sections:
- Section A: Pictorial exposition of DyRep’s representation learning module that visualizes Localized Embedding Propagation Principle, Temporal Point Process based Self-Attention and Algorithm 1.
- Section B: Discusses rationale behind DyRep framework - includes discussion on marked process view of DyRep clarifying differences of edge type vs dynamic, consolidated comparison to (Trivedi et. al. 2017) and description on support for node, edge types and unseen nodes in our framework.

Further, we have responded to individual comments below.

---

### Public Comment · (anonymous) · 2019-04-10
**Would you plan to release the source code?**

Congratulations! Would you plan to release the source code?

---

> ### Public Comment · ~Boris_Knyazev1 · 2019-09-25
> **Our implementation and extension of the paper**
>
> We tried to reimplement this method here https://github.com/uoguelph-mlrg/LDG . Please feel free to report issues or submit PRs. Our implementation is not very clean, so I hope I will find time to clean it up in the future.
>
> It's an interesting method and worth exploring further, but the presentation could be better. As a result, unfortunately, we were unable to reproduce it, even after several rounds of discussion with the authors, because there are many implementation tricks and assumptions, as well as typos in some formulas in the paper. We appreciate their detailed help though. We only tried the Social Evolution dataset and the link prediction task. It's also extremely slow to train (in our implementation), because you need to loop over all training events and it's hard to parallelize it without taking more assumptions. The Social Evolution dataset is also a challenging dataset in terms of machine learning due to its very noisy (especially Proximity events) and imbalanced events, but I guess it's a typical case in practice. We found that you can get extremely good results without any training by just computing basic statistics of events in the training set. Please see our tech report "Learning Temporal Attention in Dynamic Graphs with Bilinear Interactions" at https://arxiv.org/abs/1909.10367 for details about that, where we also show that you might not need the ground truth association graph (i.e. CloseFriends) to learn a good model. The GitHub dataset seems to be better, but we didn't evaluate on it.

---

> > ### Author Response · Authors · 2019-09-25
> > **Response**
> >
> > Thank you for your comment and efforts on studying the data and extensions of our work.
> > We will make the original implementation and data available soon. However, we want to note that
> > the data used in this extension is different in terms of statistics/pre-processing from our data and the performance doesn't seem to worsen a lot (We report results in temporal slots and performance will be slightly worse if we use single number average as done in extension).  The performance with simple stats has probably to do with high density and recurrence  (~2M in our case) in dataset  but will look in more detail.  Also, we checked the current version uploaded here and we believe it lists all assumptions and does not have any incorrect formulae. But if you point to the issue, we would be happy to rectify anything that you found incorrect.

---

> > > ### Public Comment · ~Boris_Knyazev1 · 2019-09-26
> > > **Thanks for a response!**
> > >
> > > Thanks for a response. Most of the issues were clarified to us before, and we really appreciate that. I think, however, that those clarifications should be made more accessible to other people either in the form of responses here or in the updated paper. I'll try my best to enumerate most of them here. My list applies to the Social Evolution dataset and dynamic link prediction mainly, and may or may not apply to other cases.
> > >
> > > /* I admit that some of my misunderstandings and questions come from the fact that "Temporal point processes" are still something mysterious to me. There are a lot of materials on that in the works you cite, but a more clear connection (and better, quantitative comparison) between neural networks, specifically recurrent neural networks, and point processes would be appreciated. As far as I understand, the main advantage is the support of continuous time scale, which is great. However, to me (as a neural network person), it seems to be easier to adapt an RNN to support continuous time scale. There seems to a literature on that [1]. So, it would be interesting to know your thoughts regarding that. */
> > >
> > > I've collected 11 issues/comments so far. There is, of course, no rush to address them!
> > >
> > > 1. Sign typo in Section 5.2 in the conditional density formula (should be exp(-...) as in Section 2.2). Also, there are too many inline equations, so it's hard to refer to them and it makes the reading experience more challenging.
> > > 2. Computing the exponential term in that formula in 5.2 seems to be extremely expensive, because you need to sum over all possible events between t_bar and t. It seems that you assume to sum only over events involving nodes u and v.
> > > 3. Unfortunately, most (>99%) events in the Social Dataset are symmetric Proximity events, but lambda(u,v) is not equal lambda(v,u), so it might make sense to take an average of them or something like that both during training and testing.
> > > 4. In 5.2 "Hence, when ranking the entities, we remove any entities that creates a pair already seen in the test" seems to be not applicable to the Social dataset, because it's very dense.
> > > 5. Many results are reported only on plots (e.g., Figure 2), so it's very hard to compare numerically to what we obtained.
> > > 6. The number of Association events in the test set is extremely small (~70), so I'm not sure if the comparison in Figure 2 is reliable. Also, it's not clear what the error bar actually stands for: is it std over multiple runs or time slots or something else.
> > > 7. Most of the baselines seem to be worse or on pair with the random prediction, which should be MAR~42 for 84 nodes in the dataset. This basically says that the baselines do learn anything useful. This is a little bit weird, however, we got similar results in our experiments with some static methods.
> > > 8. The Exogenous term in Eq. (4) depends on the difference between t and t_bar. It means that, for example, if you use seconds as the units, this difference can be extremely huge in some cases and the embeddings will go to NaN or be very unstable during training. I think to compute this term, it has more sense to represent t and t_bar as a vector [year, month, day, hour, minute, second, etc.] or something like that, but I'm not sure if you do that.
> > > 9. There are node features in the dataset, and the paper does not say if you use them or not.
> > > 10. Not all hyperparameters are mentioned for the DyRep model: learning rate, exact number of epochs, optimizer, weight decay, etc. Also, reporting the range of parameters is fine, as you do for baselines, but final parameters that you found to be the best is also important, because it can be very expensive to run hyperparameter search. Also, it's not clear how the search is performed, because there is no validation set.
> > > 11. The intuition behind using the log in Section 4 in the loss is not clear. Why do you need it for the first term and not for the second?
> > >
> > > [1] Zhibin Yu, Dennis S. Moirangthem, and Minho Lee2, Continuous Timescale Long-Short Term Memory Neural Network for Human Intent Understanding

---

### Meta-Review · Area_Chair1 · 2018-12-14
**Meta-Review for DyRep paper**

**Confidence:** 4
**Recommendation:** Accept (Poster)

**Metareview:**

After discussion, all reviewers agree to accept this paper. Congratulations!!